# OOD-ODBench: An Object Detection Benchmark for OOD Generalization Algorithms

## Abstract

The consensus about machine learning tasks, such as object detection, is still the test data are drawn from the same distribution as the training data, which is known as IID (Independent and Identically Distributed). However, it can not avoid being confronted with OOD (Out-of-Distribution) scenarios in real practice. It is risky to apply an object detection algorithm without figuring out its OOD generalization performance. On the other hand, a plethora of OOD generalization algorithms has been proposed to amortize the gap between the in-house and open-world performances of machine learning systems. However, their effectiveness was only demonstrated in the image classification tasks. It is still an opening question of how these algorithms perform on more complex tasks. In this paper, we first specify the setting of OOD-OD (OOD generalization object detection). Then, we propose OOD-ODBench consisting of four OOD-OD benchmark datasets to evaluate various object detection and OOD generalization algorithms. From extensive experiments on OOD-ODBench, we find that existing OOD generalization algorithms fail dramatically when applied to the more complex object detection tasks. This raises questions over the current progress on a large number of these algorithms and whether they can be effective in practice beyond simple toy examples. For future work, we sincerely hope that OOD-ODBench can serve as a foothold for OOD generalization object detection research.

## 1 Introduction

Modern object detection methods (Liu et al., 2021; Huang et al., 2019; Pang et al., 2019; Wu et al., 2019; Zhang et al., 2020a; Sun et al., 2020; Zhu et al., 2021; Ge et al., 2021) have achieved many progresses on various applications, such as autonomous driving and industrial defect detection. Tremendous efforts have been devoted to improving an object detector's performance on standard datasets, such as MS-COCO (Lin et al., 2014). While these efforts have seen impacts on industry (Redmon et al., 2016; Redmon & Farhadi, 2017; 2018; Bochkovskiy et al., 2020; Ge et al., 2021), the improvements are becoming marginal recently and most achievements are accompanied by an inherent assumption, *i.e.*, the training data and the test data are IID (Independent and Identically Distributed). However, this assumption is unlikely to hold in real-world scenarios. For example, an autonomous system suffers from different environmental conditions (Dai & Gool, 2018; Volk et al., 2019); a medical system fails to work consistently among hospitals when data are collected from different equipment (de Castro et al., 2019; Albadawy et al., 2018; Perone et al., 2019). As a consequence, models trained on IID dataset are susceptible to a subtle disturbance in test data distribution (Out-of-Distribution) and fail to generalize to real scenarios (Torralba & Efros, 2011). Previous research devoted to encountering this train-test discrepancy can be summarized as either *"less complex"* or *"complex but not general"*. From the first perspective, a plethora of Domain Generalization (DG) algorithms (Arjovsky et al., 2019; Ahuja et al., 2021; Li et al., 2018b; Sun & Saenko, 2016; Xu et al., 2020c; Yan et al., 2020; Krueger et al., 2021; Pezeshki et al., 2020; Parascandolo et al., 2021; Koyama & Yamaguchi, 2021; Huang et al., 2020; Sagawa et al., 2019) concentrate on improving OOD generalization ability. But they are simply evaluated on the image classification. The effectiveness is unknown when applied to the complex task (object detection). On the other perspective, numerous Domain Adaption (DA) algorithms (Chen et al., 2018; He & Zhang, 2020; Rodriguez & Mikolajczyk, 2019; Xu et al., 2020a; Su et al., 2020; Xu et al., 2020b; Soviany et al., 2019; Deng et al., 2020; Chen et al., 2021) aim to build an optimal object detector that can be generalized into

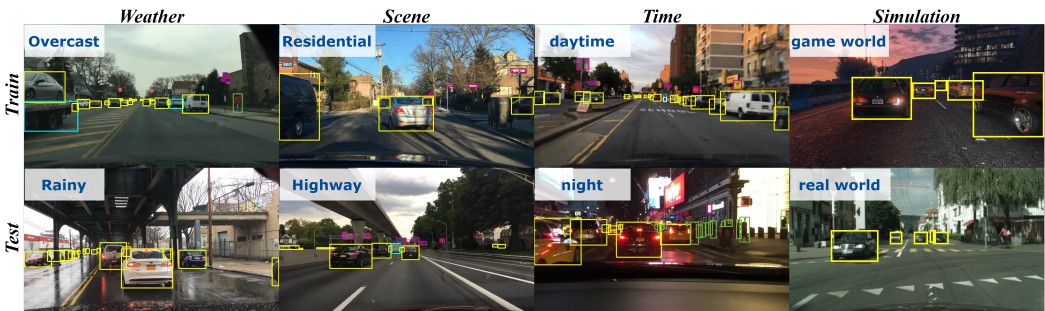

Figure 1: The setting illustration of out-of-distribution generalization object detection (OOD-OD).

a pre-specified target domain. However, it is hard to ensure performance consistency when dealing with unseen and infinite real-world domains. In this paper, we focus on **OOD generalization object detection (OOD-OD)** which aims at training detectors to generalize to the testing data drawn from an unseen distribution distinct from the training distribution. See Table 1 for more details and we provide the theoretical definition for OOD-OD in Appendix A.1.

In this work, we propose OOD-ODBench, in which four OOD-OD benchmarks are constructed with existing datasets, including BDD100K (Yu et al., 2018), Cityscapes (Cordts et al., 2016) and Sim10K (Johnson-Roberson et al., 2016). As revealed by (Ye et al., 2021), data distribution shifts on classification datasets are dominated by correlation shift and diversity shift. We test whether a similar phenomenon also exists on detection datasets and we construct a synthetic dataset named CtrlShift to quantitatively analyze generalization ability over the two kinds of distribution shifts of OOD-OD, respectively. With the above benchmark datasets, numerous experiments are conducted with detectors ranging from one(two)-stage to transformer-based and diverse OOD generalization algorithms carefully implemented on popular detectors (Ren et al., 2015; Lin et al., 2017b).

Section 2 reviews the related work dispersed in different research areas. Section 3 provides clarification of different techniques and tasks with similar names. Section 4 introduces the implementation details of OOD-ODBench, including the datasets, algorithms, and model selection methods. Finally, Section 5 discusses the experiment results on OOD-ODBench and offers insightful recommendations for future work. Our main contributions can be summarized as followed:

1. We propose OOD-ODBench, the first OOD generalization benchmark for object detection algorithms. Based on the extensive experiment results, we arrive at a surprising conclusion: *The enormous achievements in IID object detection are marginal on OOD generalization object detection, and the OOD generalization improvements on classification are hard to generalize to more complex tasks (i.e., object detection).*

2. In OOD-ODBench, we propose a Sim2real benchmark for OOD generalization object detection analysis which measures the possibility of training models with low-cost simulated data to generalize well on real scenarios.

3. To further analyze the generalization ability under the different types of shifts, we construct a synthetic dataset with designed shifts, namely CtrlShift. The synthetic dataset can systematically measure the OOD generalization algorithms' performances under different types of distribution shifts.

4. From Benchmark results and analysis, we recommend future research to clearly investigate the diversity shift and the correlation shift on OOD generalization object detection before designing algorithms and then evaluate them comprehensively on the two-dimensional shift using CtrlShift.

## 2 RELATED WORK

### 2.1 OBJECT DETECTION

The task of object detection aims at classifying and localizing the objects in an image based on the assumption that test data are drawn from the same distribution as training data. Modern deep

Table 1: Comparison between OOD generalization object detection and other tasks. $X^i, Y^i$ indicate the data drawn from distribution $i$ and $Y_1, Y_2$ represent category labels and bounding box labels respectively.

| Setup | Task | Training inputs | Test inputs | Outputs |
|---|---|---|---|---|
| Unsupervised learning | classify / detect | $X^1$ | $X^1$ | $Y^1$ |
| Supervised learning | classify / detect | $X^1, Y^1$ | $X^1$ | $Y^1$ |
| Semi-supervised learning | classify / detect | $X^1, (Y^1)'$ | $X^1$ | $Y^1$ |
| Transfer learning | classify / detect | $X^{1,...,d}, X^{d+1}, Y^{d+1}$ | $X^{d+1}$ | $Y^{d+1}$ |
| Domain generalization | classify | $X^{1,...,d}, Y^{1,...,d}$ | $X^{d+1}$ | $Y_1^{d+1}$ |
| Domain adaption | detect | $X^{1,...,d}, Y^{1,...,d}, X^{d+1}$ | $X^{d+1}$ | $Y_{1,2}^{d+1}$ |
| OOD-OD | detect | $X^{1,...,d}, Y^{1,...,d}$ | $X^{d+1,...}$ | $Y_{1,2}^{d+1,...}$ |

learning-based object detection models can be divided into three categories: two-stage detectors (Girshick et al., 2014; Grauman & Darrell, 2005; Girshick, 2015; Ren et al., 2015; Lin et al., 2017a; Dai et al., 2016; He et al., 2017; Qiao et al., 2021; Cai & Vasconcelos, 2019; Huang et al., 2019; Pang et al., 2019; Wu et al., 2019; Sun et al., 2020), one-stage detectors (Redmon et al., 2016; Redmon & Farhadi, 2017; 2018; Bochkovskiy et al., 2020; Ge et al., 2021; Liu et al., 2016; Lin et al., 2017b; Zhou et al., 2019; Tan et al., 2020; Law & Deng, 2018; Tian et al., 2019; Zhang et al., 2020a; Zhu et al., 2021; Liu et al., 2021) and lightweight detectors with small components (Howard et al., 2017; Sandler et al., 2018; Howard et al., 2019; Zhang et al., 2018; Ma et al., 2018; Wang et al., 2018; Iandola et al., 2016). Compared to one-stage detectors, two-stage detectors are equipped with a separate differentiable module to generate region proposals which are possible to contain objects. Lightweight detectors are usually proposed to improve real-time performance with a small and efficient network. Recently, with the enormous success of applying transformer (Vaswani et al., 2017) on computer vision, a branch of transformer-based detector (Zhu et al., 2021; Liu et al., 2021) has shaped up.

## 2.2 OOD GENERALIZATION

The task of OOD generalization is training on multiple datasets sampled from distinct domains and then generalizing to an unseen test domain. Models with OOD generalization ability typically have access to multiple training datasets for the same task obtained from various environments. The purpose of OOD generalization algorithms is to learn from these diverse but relevant training settings before being applied to unknown testing environments. Driven by this motivation, many algorithms have been proposed throughout these years. These algorithms can be divided into: empirical risk learning (Vapnik, 1998; Sagawa et al., 2019), invariant risk optimization (Arjovsky et al., 2019), domain adversarial learning (Ajakan et al., 2014; Li et al., 2018c; Ruan et al., 2021), meta-learning (Zhang et al., 2020b; Li et al., 2018a), kernel function (Li et al., 2018b; Sun & Saenko, 2016), gradient-based approach (Shi et al., 2021; Pezeshki et al., 2020; Bai et al., 2020; Parascandolo et al., 2021; Shahtalebi et al., 2021; Koyama & Yamaguchi, 2021; Rame et al., 2021), risk extrapolation (Krueger et al., 2021), data processing (Xu et al., 2020c; Yan et al., 2020), transfer learning (Blanchard et al., 2017; Xu & Jaakkola, 2021), information bottleneck (Ahuja et al., 2021) and self-supervised learning (Wang et al., 2020; Zhou et al., 2020).

OOD generalization for object detection is currently underexplored. Region Aware Proposal reweighTing (RAPT) (Zhang et al., 2022) aims to eliminate dependence within RoI features for domain generalization. Cyclic-Disentangled Self-Distillation (Wu & Deng, 2022) aims at disentangling domain-invariant representations. 3D-VField (Lehner et al., 2022) improves generalization on 3D object detection.

## 2.3 OOD BENCHMARK

Different domains data (Zhou et al., 2021; Wang et al., 2022) can be viewed as data drawn from different distributions and the distinct train-test domains are Out-of-Distribution. DomainBed (Gulrajani & Lopez-Paz, 2020) is a large-scale benchmark suite for reproducing domain generalization research and facilitating the implementation of new algorithms. With the experiment results of fourteen algorithms on seven datasets, the authors found that empirical risk minimization (Vapnik, 1998)

Table 2: Details of datasets used in benchmarks. **Quantity** indicates the number of images in each domain. **Total** counts the total number of training and testing domains respectively.

| Benchmark | Dataset | Domain | Train | Test | Quantity | Total |
|---|---|---|:---:|:---:|---|---|
| Weather | BDD100K | clear | √ | | 42690 | 52699 |
| | | overcast | √ | | 10009 | |
| | | foggy | | √ | 143 | |
| | | cloudy | | √ | 5619 | 17888 |
| | | rainy | | √ | 5808 | |
| | | snowy | | √ | 6318 | |
| Scene | BDD100K | city street | √ | | 49628 | 69506 |
| | | highway | √ | | 19878 | |
| | | gas station | | √ | 34 | |
| | | parking lot | | √ | 426 | 9943 |
| | | residential | | √ | 9327 | |
| | | tunnel | | √ | 156 | |
| Time | BDD100K | daytime | √ | | 41986 | 47791 |
| | | dawn dusk | √ | | 5805 | |
| | | night | | √ | 31900 | 31900 |
| Sim2real | Sim10K | simulation | √ | | 8500 | 8500 |
| | Cityscapes | reality | | √ | 3975 | 3975 |

achieves state-of-the-art performance across all datasets. OoD-Bench (Ye et al., 2021) identifies and measures two distinct kinds of distribution shifts in various OOD datasets. With tremendous empirical learning results, the authors recommend that algorithms should be comprehensively evaluated on two types of datasets dominated by correlation shift and diversity shift respectively.

## 3 CLARIFICATION OF TASKS

**Domain Randomization** techniques (Tobin et al., 2017; Tremblay et al., 2018; Zakharov et al., 2019; Yue et al., 2019; Huang et al., 2021) aim at providing enough simulated domains at training data so that models are possible to generalize to real-world scenarios based on a hypothesis that with enough variability in the data simulator, the real world may appear to be a specific variation of the simulation data which exists in the training set.

**OOD Detection for Object Detection** (Joseph et al., 2021; Du et al., 2022a; Harakeh & Waslander, 2021; Riedlinger et al., 2021; Dhamija et al., 2020; Miller et al., 2019; Hall et al., 2020; Deepshikha et al., 2021) can be formulated as a binary classification problem which distinguishes whether the distribution of the incoming data is out of the distribution of the training data.

**Open-World Object Detection** (Joseph et al., 2021; Zhao et al., 2022) initially learns a model which can detect all the previously encountered categories, and incrementally updates the model when unseen classes come.

**Open-Vocabulary Object Detection** (Gu et al., 2021; Zareian et al., 2021; Du et al., 2022b; Bravo et al., 2022) aims to train an detector which can detect various objects in any novel categories described by arbitrary texts.

## 4 OOD-ODBENCH: IMPLEMENTATION DETAILS

### 4.1 BENCHMARKING DATASETS

In OOD-ODBench, we choose datasets to cover as many types of variations between training and test datasets as possible. Figure 2 lists some samples of the four benchmark datasets.

**BDD100K** (Yu et al., 2018) contains 80,000 labeled images (70,000 for training and 10,000 for validation) with ten annotated object categories, including bike, bus, car, motor, person, rider, traffic light, traffic sign, train and truck. Each image has three attribute labels which indicate the condition, including the weather, scene and time for data collection and we remove the images with an undefined attribute label. Specifically, we construct three OOD environments using the attribute labels, including **Weather**, **Scene**, and **Time**.

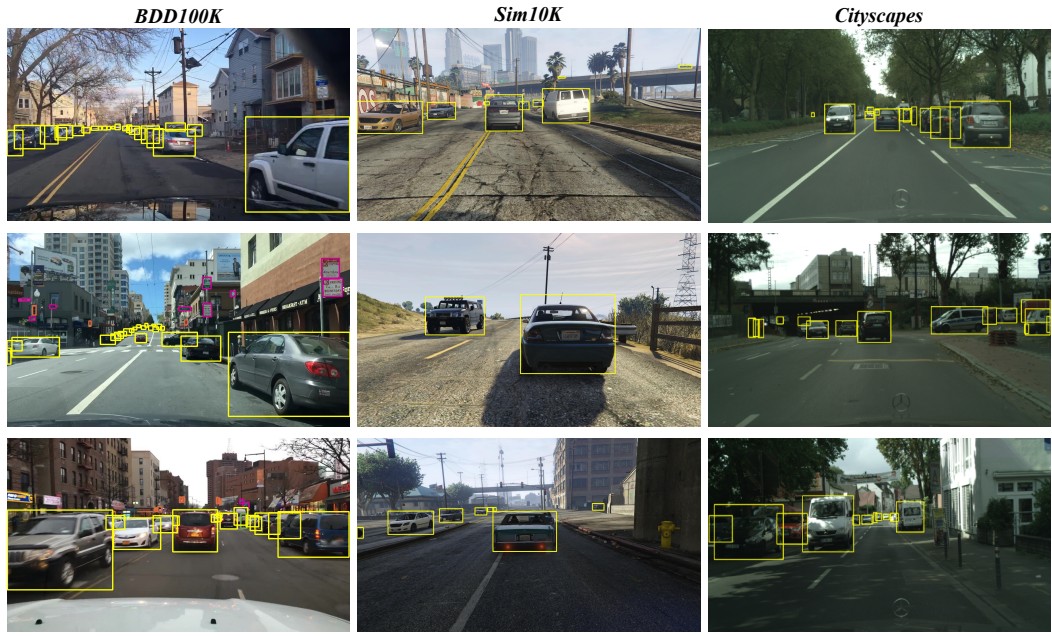

Figure 2: Some samples of datasets included in OOD-ODBench. Note that BDD100K has ten categories while Sim10K and Cityscapes only use the annotated cars.

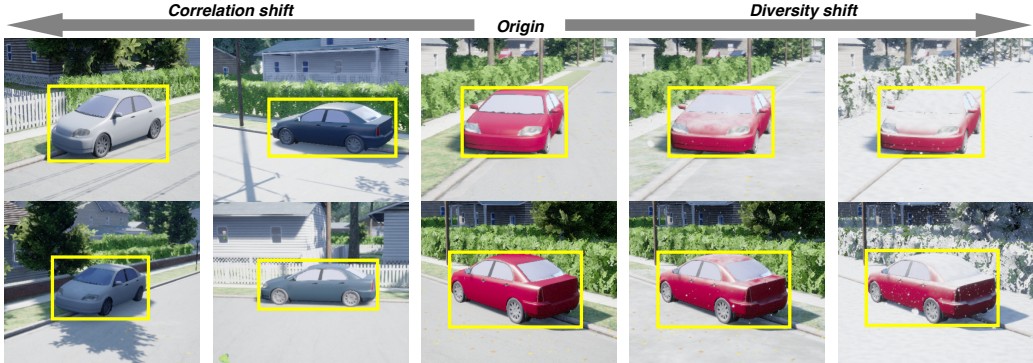

Figure 3: Some samples of CtrlShift and the illustration of the two-dimensional shift.

**Sim10K** (Johnson-Roberson et al., 2016) is a synthetic dataset containing 10,000 images (8,000 for training, 1,000 for validating and 1,000 for testing) with bounding box annotations for cars, which is rendered with the Grand Theft Auto V (GTA5) game engine.

**Cityscapes** (Cordts et al., 2016) is a large-scale database which focuses on urban street scenes. The dataset consists of around 5000 fine annotated images (2975 for training, 500 for validating and the rest for testing) with eight annotated instance categories. On OOD-ODBench, we consider the car recognition task to construct the Sim2real benchmark for simplicity and without loss of generality. We construct the **Sim2Real** benchmark which covers the "sim2real" scenario. The simulated images of Sim10K are used for training and the real images of Cityscapes are used for testing.

**CtrlShift** is a synthetic dataset to analyze the two-dimension shift in OOD generalization object detection. The Airsim simulator (Shah et al., 2017) based on the high-fidelity rendering software Unreal Engine 4 is used to generate samples in CtrlShift. We totally sample over 2000 simulated images from both the rural and urban environments which contain common objects, including buildings, traffic lights, and vehicles. Moreover, every image comprises two attribute labels to construct a controllable distribution shift. One is car color which indicates the color of the car in the image,

Table 3: Experimental results of detectors performance measured by AP(%) on MS COCO and the four OOD benchmarks of OOD-ODBench. All models are implemented by mmdetection (Chen et al., 2019) and loaded pretrained weights provided by open-mmlab (Contributors, 2018). Note that @Faster in Libra R-CNN row represents applying Faster R-CNN as architecture. X-101-64x4d represents a modified ResNeXt-101 network architecture from (Xie et al., 2017), R-50 and R-101 represents ResNet backbones with 50 or 101 layers (He et al., 2016). OOD_Avg calculate the average accuracy on the four OOD benchmark.

| Detector | Backbone | MS COCO | Sim2real | Weather | Scene | Time | OOD_Avg |
|---|---|---|---|---|---|---|---|
| Faster R-CNN | X-101-64x4d | 42.1 | 52.1 | 35.3 | 36.0 | 26.4 | 37.5 |
| RetinaNet | X-101-64x4d | 41.0 | 49.3 | 34.7 | 35.8 | 24.6 | 36.1 |
| Mask R-CNN | X-101-64x4d | 42.8 | 52.9 | 34.8 | 35.8 | 26.8 | 37.6 |
| CornetNet | Hourglass104 | 41.2 | 27.8 | 29.7 | 30.8 | 22.5 | 27.7 |
| YOLOv3 | DarkNet-53 | 33.7 | 38.2 | 27.0 | 28.2 | 19.2 | 28.2 |
| FCOS | X-101-64x4d | 42.6 | 50.8 | 35.6 | 36.5 | 24.8 | 36.9 |
| Cascade R-CNN | X-101-64x4d | 44.7 | 53.1 | 36.3 | 36.7 | 24.1 | 37.6 |
| MS R-CNN | X101-64x4d | 43.0 | 52.5 | 35.5 | 35.7 | 25.5 | 37.3 |
| Libra R-CNN@Faster | X-101-64x4d | 42.7 | 51.4 | 34.9 | 35.5 | 25.5 | 36.8 |
| Double-Head R-CNN | R-50 | 40.0 | 52.2 | 35.3 | 35.2 | 25.5 | 37.1 |
| VarifocalNet | X-101-64x4d | 50.4 | 56.1 | 38.7 | 39.0 | 27.4 | 40.3 |
| Sparse R-CNN | R-101 | 46.2 | 54.1 | 36.3 | 36.4 | 28.2 | 38.8 |
| DETR | R-50 | 42.0 | 37.8 | 23.5 | 24.4 | 15.8 | 25.4 |
| Deformable DETR | R-50 | 46.8 | 53.3 | 35.1 | 35.3 | 26.1 | 37.5 |
| Swin Transformer | Swin-B | 51.9 | 58.4 | 36.6 | 32.0 | 28.9 | 39.0 |
| YOLOX | YOLOX-x | 50.9 | 51.1 | 33.1 | 34.8 | 29.4 | 37.1 |

the other is snow intensity which indicates the intensity of the weather snow added using Airsim's plugin. Cars are annotated in the dataset since the car is a common object in existing datasets. To construct CtrlShift dominated by correlation shift, we restrict the white cars only existing in the training set and control the quantity ratio $\rho_{correlation}$ of white car data in the training set. In a way that spuriously correlates strongly with the class label since the color white will be more relevant to the car label in the training set when the $\rho_{correlation}$ increases, which increases the correlation shifts in the dataset. And for CtrlShift dominated by diversity shift, we render snow weather effects on each training datum with a certain intensity $\rho_{diversity}$ and the increase of $\rho_{diversity}$ corresponds to a larger diversity shift. See Figure 3 for some examples. Moreover, we provide an API to generate the training set and the testing set with specific choices of $\rho_{diversity}$ and $\rho_{correlation}$.

## 4.2 DETECTION METHODS FOR COMPARISONS

We choose widely-used detectors trained with the empirical risk minimization (ERM) or OOD generalization algorithms on OOD-ODBench. The comparisons of different detectors trained with ERM on OOD-ODBench can help answer the question that whether the progress made by recently proposed detectors is generalizable to OOD data. The benchmarks of detectors trained with proposed OOD generalization algorithms can indicate whether the OOD generalization algorithms proposed recently are still effective for object detection beyond toy image classification tasks.

**Detectors.** Object detection models (detectors) generally can be categorized into two genres: one-stage methods and two-stage methods. One-stage detectors predict the bounding boxes as well as the categories of the objects. Two-stage detectors predict the bounding boxes first to indicate the possible locations of objects. Then, two-stage detectors conduct classifications on the images within the bounding boxes to predict the categories of these images. Recently, with the tremendous success of transformer (Vaswani et al., 2017), transformer-based detectors become popular. We have selected one/two-stage and transformer-based algorithms ranging from 2015 to 2021 for our Object Detection OOD generalization benchmark. One-stage detectors: **RetinaNet** (Lin et al., 2017b), **CornerNet** (Law & Deng, 2018), **YOLOv3** (Redmon & Farhadi, 2018), **FCOS** (Tian et al., 2019), **VarifocalNet** (Zhang et al., 2020a), **YOLOX** (Ge et al., 2021). Two-stage detectors: **Faster R-CNN** (Ren et al., 2015), **Mask R-CNN** (He et al., 2017), **Cascade R-CNN** (Cai & Vasconcelos, 2019), Mask Scoring R-CNN (**MS R-CNN**) (Huang et al., 2019), **Libra R-CNN** (Pang et al., 2019), **Double-Head R-CNN** (Wu et al., 2019) and **Sparse R-CNN** (Sun et al., 2020). Transformer-based detectors: **DETR**

Table 4: Experimental results of domain generalization algorithms on OOD generalization object detection. All algorithms are implemented by ourselves based on Faster R-CNN (Girshick, 2015) with ResNet-50 (He et al., 2016) backbone. Hyper-parameter of each algorithm is chosen among 0.1, 1 and 10 according to the average AP over the four benchmarks.

| Algorithm | Hyper-parameters | Sim2real | Weather | Scene | Time | Average |
|---|---|---|---|---|---|---|
| ERM | - | 44.1 | 24.4 | 24.1 | 17.7 | 27.6 |
| IB-ERM | $\lambda_{ib} = 0.1$ | 42.5 | 24.7 | 23.4 | 18.2 | 27.2 |
| IRM | $\lambda_{irm} = 0.1$ | 42.3 | 24.9 | 23.2 | 17.5 | 27.0 |
| MMD | $\gamma_{mmd} = 1$ | 42.9 | 24.1 | 23.9 | 18.4 | 27.3 |
| CORAL | $\gamma_{mmd} = 10$ | 42.6 | 24.4 | 22.8 | 17.7 | 26.9 |
| VREx | $\lambda_{vrex} = 1$ | 43.1 | 24.7 | 24.3 | 18.0 | 27.5 |
| GS | $\lambda_{reg} = 10$ | 40.3 | 19.9 | 18.4 | 15.2 | 23.5 |
| IGA | $\lambda_{penalty} = 1$ | 42.9 | 24.6 | 24.3 | 18.6 | 27.6 |
| GroupDRO | $\eta_{groupdro} = 1$ | 42.8 | 24.5 | 23.6 | 18.1 | 27.3 |
| RSC | $\lambda_{rsc} = 10$ | 41.0 | 8.7 | 8.8 | 7.0 | 16.4 |
| CAD | $\lambda_{cad} = 1$ | 42.8 | 24.2 | 23.1 | 18.4 | 27.1 |
| CausIRL | $\lambda_{cirl} = 1$ | 42.4 | 23.8 | 23.4 | 17.1 | 26.7 |

(Carion et al., 2020), **Deformable DETR** (Zhu et al., 2021) and **Swin Transformer** (Liu et al., 2021).

**OOD generalization algorithms.** We have adapted eleven algorithms from different OOD research areas to the classification branch in object detection, including Empirical Risk Minimization (**ERM**) (Vapnik, 1998) which aims to minimize the loss function overall the training domains, (**IB-ERM**) (Ahuja et al., 2021) which applies an information bottleneck constraint to address OOD generalization, Invariant Risk Minimization (**IRM**) (Arjovsky et al., 2019) which aims at estimating invariant correlations across different domains, adversarial feature learning (**MMD**) (Li et al., 2018b) which imposes Maximum Mean Discrepancy (Gretton et al., 2012) to align the distributions among different domains, correlation alignment (**CORAL**) (Sun & Saenko, 2016) which aims at matching the mean and covariance of feature distributions, Variance Risk Extrapolation (**VREx**) (Krueger et al., 2021) which performs a form of robust optimization over extrapolated domains, Gradient Starvation (**GS**) (Pezeshki et al., 2020) which derives a regularization to overcome the gradient descent phenomenon across different domains, (**IGA**) (Koyama & Yamaguchi, 2021) which uses a parametrization trick to conduct feature searching and predictor training, Group Distributionally Robust Optimization (**GroupDRO**) (Sagawa et al., 2019) which increases the importance of each domain with penalty loss, and Representation Self-Challenging (**RSC**) (Huang et al., 2020) iteratively challenges the dominant features to force the model to activate the remaining features. Optimal Representations (**CAD**) (Ruan et al., 2021) designs self-supervised objectives to obtain representations on which risk is minimal to any distribution. Invariant Causal Mechanisms (**CausIRL**) (Chevalley et al., 2022) learns the invariant features by viewing the learning process as a causal process and introduces a unifying framework.

### 4.3 Model Selection Methods

Model selection methods can influence the final rankings of methods to a large extent, especially in OOD generalization tasks (Gulrajani & Lopez-Paz, 2020). However, there is no consensus on what parameters selection strategy should be used in OOD generalization research for object detection. In OOD-ODBench, we choose the models trained at the last epoch as our model selection method. This is because the testing data is inaccessible and selecting models based on the trainset's performances may lead to excessive over-fitting for current methods since there is a huge distribution gap between the training set and testing set. For future research, we strongly recommend that researchers should detail and include the model selection methods in OOD generalization object detection research.

## 5 Experiment Results

### 5.1 Benchmark Results

Table 5: Ablation study of OOD algorithms with different detectors on Sim2real. Faster R-CNN (Ren et al., 2015), RetinaNet (Lin et al., 2017b) and DETR (Carion et al., 2020) are all with ResNet-50 (He et al., 2016) backbone.

| Algorithm | Hyper-parameters | Faster R-CNN | RetinaNet | DETR | Average |
|---|---|---|---|---|---|
| ERM | - | 44.1 | 38.5 | 36.0 | 39.5 |
| IB-ERM | $\lambda_{ib} = 0.1$ | 42.5 | 38.8 | 35.1 | 38.8 |
| IRM | $\lambda_{irm} = 0.1$ | 42.3 | 38.4 | 34.7 | 38.5 |
| MMD | $\gamma_{mmd} = 1$ | 42.9 | 14.0 | 36.2 | 31.0 |
| CORAL | $\gamma_{mmd} = 10$ | 42.6 | 35.8 | 36.9 | 38.4 |
| VREx | $\lambda_{vrex} = 1$ | 43.1 | 37.6 | 36.9 | 39.2 |
| GS | $\lambda_{reg} = 10$ | 40.3 | 33.0 | 33.0 | 35.4 |
| IGA | $\lambda_{penalty} = 1$ | 42.9 | 38.2 | 29.5 | 36.9 |
| GroupDRO | $\eta_{groupdro} = 1$ | 42.8 | 37.3 | 36.2 | 38.8 |
| RSC | $\lambda_{rsc} = 10$ | 41.0 | 38.4 | 36.9 | 38.8 |
| CAD | $\lambda_{cad} = 1$ | 42.8 | 37.9 | 35.5 | 38.7 |
| CausIRL | $\lambda_{cirl} = 1$ | 42.4 | 38.3 | 36.3 | 39.0 |

In this section, we conduct numerical experiments on our benchmark to reveal the OOD generalization ability for existing algorithms and we provide further discussion in Appendix A.5. All experiments are conducted on a Pytorch platform with eight Tesla V100 GPUs. We evaluate each algorithm using the Average Precision (AP) from MS COCO (Lin et al., 2014). For object detection algorithms, our codes are based on mmdetection (Chen et al., 2019) and for domain generalization algorithms, our codes are stemmed from DomainBed (Gulrajani & Lopez-Paz, 2020). We draw several conclusions from the results.

**The enormous achievements of object detection on IID datasets are marginal on the OOD condition.** Table 3 summarizes the OOD results of various object detection algorithms. If we use the classic Faster R-CNN (Ren et al., 2015) as our baseline, all successful algorithms on object detection only marginally improve OOD generalization ability by 2.8 AP (Varifo-

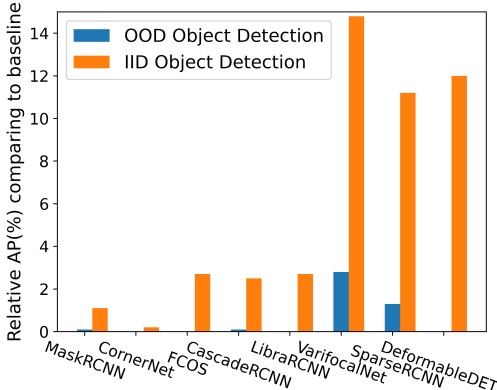

Figure 4: The improvements of recent object detection methods over the baseline on IID and OOD respectively. While the improvements on IID datasets (MS COCO) are prominent, it is not generalizable to OOD scenarios. The compared baseline method is Faster R-CNN.

calNet (Zhang et al., 2020a)) overall the four OOD benchmarks. While on the IID object detection benchmark (MS COCO test-dev (Lin et al., 2014)), this performance improvement can be up to 14.8 AP comparing to VarifocalNet (Zhang et al., 2020a). Figure 4 intuitively displaces the significant discrepancy between IID and OOD. What is responsible for these results? We suspect two factors: One is that current researches simply stem from the ideal assumption of IID regardless of whether it can be satisfied in real scenarios. The other is that the improvement on IID datasets may be a phenomenon of over-fitting since few works provide sufficient evidence that the causal features have been learned during the training process without evaluating on OOD benchmarks.

**The tremendous success of domain generalization algorithms confronting OOD is inconsistent between classification and object detection.** We draw this conclusion from Table 4 and the experimental results reported on OOD-bench (Ye et al., 2021). The OOD results on the four benchmarks in Table 4 suggest that the domain generalization algorithms degenerate or slightly outperform the ERM (Vapnik, 1998) which can be attributed to the hyper-parameters bias. Moreover, as for VREx (Krueger et al., 2021) which is the best models on Correlation-Bench (Ye et al., 2021), AP drops by 0.1 comparing to ERM (Vapnik, 1998) while VREx (Krueger et al., 2021) outperforms ERM (Vapnik, 1998) by 8.6 AP on Correlation-Bench (Ye et al., 2021). RSC (Huang et al., 2020) which is the

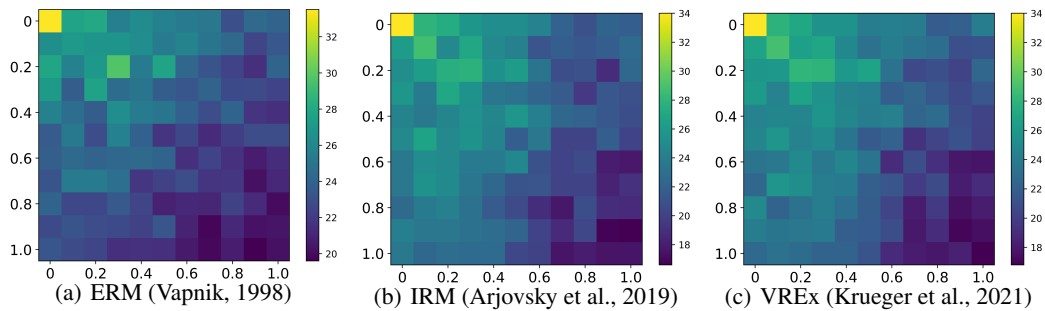

Figure 5: Controlled distribution shifts experiments results of ERM, IRM and VREx. X-axis is Diversity shift and Y-axis is Correlation shift. Each block indicates the AP(%).

best models on Diversity-Bench (Ye et al., 2021) degenerates 11.2 AP while improves 0.6 accuracy comparing to ERM (Vapnik, 1998).

**The generalization inconsistency between classification and object detection of domain generalization algorithms happens among different detectors.** As shown in Table 4.2, we choose the popular two-stage detector, Faster R-CNN (Ren et al., 2015), one-stage detector, RetinaNet (Lin et al., 2017b), and transformer-based detector, DETR (Carion et al., 2020), as base models for implementing domain generalization algorithms. Obviously, ERM achieves the best average generalization ability on the three detectors and we can conclude that the degeneration of domain generalization algorithms has little relevance to the detectors.

## 5.2 CONTROLLED DISTRIBUTION SHIFTS EXPERIMENTS

Previous experiments provide performance analysis on the real scenarios for OOD generalization object detection. But it is hard to see which kind of distribution shift leads to performance degeneration. To systematically analyze the generalization performance under the influence of the two-dimensional distribution shift, we test the performance of Faster-RCNN trained by ERM (Vapnik, 1998), and the top performers on previous datasets (IRM (Arjovsky et al., 2019) and VREx (Krueger et al., 2021)) on CtrlShift dataset with different settings of the correlation shift and diversity shift. The results are shown in Figure 5, all methods consistently (Vapnik, 1998; Arjovsky et al., 2019; Krueger et al., 2021) achieve the best AP when both correlation shift and diversity shift are low. For ERM (Vapnik, 1998), the performance evenly degenerates on two dimensions. In Figure 5(b), with the increase of the two-dimension shift, the performance of IRM (Arjovsky et al., 2019) in the horizontal direction tends to degenerate faster than in the vertical direction. This indicates that IRM (Arjovsky et al., 2019) confronts correlation shift better than diversity shift. From Figure 5(c), we can observe the similar phenomenon happens for VREx (Krueger et al., 2021) on the two-dimensional shift. This phenomenon demonstrates that existing OOD generalization algorithms may help mitigate performance degradation when confronted with correlation shifts. Whereas for diversity shift, key components are missing to improve the generalization abilities, let alone the complex mixture of both shifts in real datasets. For future research, we recommend that both shifts should be included in new benchmark datasets and algorithms should be evaluated on both types of distribution shifts simultaneously.

## 6 CONCLUSION AND DISCUSSION

In this paper, we propose the first benchmark for OOD-OD tasks, named OOD-ODBench. The benchmark suite includes four benchmark datasets along with a synthetic dataset to generate controlled distribution shifts. The experimental results conducted on OOD-ODBench suggest that the enormous achievements in classical IID object detection are marginal on OOD generalization object detection. And the OOD generalization methods mainly tested on classification cannot generalize to object detection tasks. This raises questions about existing progress on object detection and OOD generalization algorithms. We appeal for more attention from the community for this problem to propose an OOD-OD method that is undoubtedly effective.

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

# A APPENDIX

## A.1 THEORETICAL ANALYSIS OF OOD GENERALIZATION OBJECT DETECTION

The OOD generalization object detection has been fragmentally researched in previous research, however, no rigorous definition of OOD generalization object detection has been given. We give a definition and taxonomy as follows:

**OOD generalization object detection (OOD-OD):** In object detection tasks, algorithms learn a mapping function $f$ to predict the category ($\mathbf{y}$) and location of interested objects in an image $\mathbf{x}$. In OOD generalization object detection tasks, training and test data pairs ($\mathbf{X}, \mathbf{Y}$) are not necessarily drawn from the same distribution. This poses great challenges for existing machine learning methods as most methods are reliant on exploiting the correlation between $\mathbf{X}$ and $\mathbf{Y}$. Due to the distribution change, the correlation might not be generalizable. More specifically, we depict the causal data generating process in Figure 6. When the data $\mathbf{X}$ are given, the causal features $\mathbf{Z}_1$ and the non-causal features $\mathbf{Z}_2$ are given. The causal

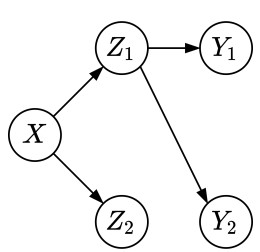

Figure 6: The causal influence among the concerned variables.

features reliably determine the location and categories of interested objects in the input images. The non-causal features are irrelevant features for predictions. An intuitive example is that if we want to recognize a dog in an image, the causal features are dogs' shapes. The non-causal features are the environment features, such as weather or captured time of the image. We improve the definitions in (Ye et al., 2021) and propose the following mathematical definitions for $\mathbf{Z}_1$ and $\mathbf{Z}_2$ given overall semantic features $\mathbf{z}$[1]:

$$\forall z \in \mathbf{Z}_1 : p(z) \cdot q(z) \neq 0 \land \forall y_1 \in \mathbf{Y}_1 : p(y_1|z) = q(y_1|z) \land \forall y_2 \in \mathbf{Y}_2 : p(y_2|z) = q(y_2|z) \quad (1)$$

$$\exists z \in \mathbf{Z}_2 : p(z) \cdot q(z) = 0 \lor \exists y_1 \in \mathbf{Y}_1 : p(y_1|z) \neq q(y_1|z) \lor \exists y_2 \in \mathbf{Y}_2 : p(y_2|z) \neq q(y_2|z) \quad (2)$$

where $p$ is the training distribution and $q$ is the test distribution. Since $\mathbf{Z}_1$ is the stable and reliable predictor for the category and location of objects, there are two kinds of shifts intuitively because of the discrepancy of $\mathbf{Z}_2$ in training and test distribution. Diversity shift stems from the first kind of features in $\mathbf{Z}_2$ since the diversity of data is embodied by novel features not shared by the environments; whereas correlation shift is caused by the second kind of features in $\mathbf{Z}_2$ which is spuriously correlated with some $\mathbf{Y}_1$ or $\mathbf{Y}_2$. Based on this, we partition $\mathbf{Z}_2$ into two subsets:

$$\mathbf{S} := \{\mathbf{z} \in \mathbf{Z}_2 | p(\mathbf{z}) \cdot q(\mathbf{z}) = 0\} \quad \mathbf{T} := \{\mathbf{z} \in \mathbf{Z}_2 | p(\mathbf{z}) \cdot q(\mathbf{z}) \neq 0\} \quad (3)$$

**Definition A.1.** *(Definition of Diversity shift and Correlation shift for OOD-OD.) Given $\mathbf{S}$ and $\mathbf{T}$ defined in Equation 3, the definitions of diversity shift and correlation shift are given as follows:*

$$D_{diversity} := \frac{1}{2} \int_{\mathbf{S}} |p(z) - q(z)| \, dz \quad (4)$$

$$D_{correlation} := \frac{1}{2} \int_{\mathbf{T}} \sqrt{p(z) \cdot q(z)} \int_{y_2} \sum_{y_1} |p(y_1, y_2|z) - q(y_1, y_2|z)| \, dz \quad (5)$$

It can be seen that both $D_{\text{diversity}}$ and $D_{\text{correlation}}$ are within the range of $[0, 1]$. $D_{\text{diversity}}$ measures the support difference non-causal features for object detection. While $D_{\text{correlation}}$ gauges the variations of conditional probabilities of the object category $\mathbf{Y}_1$ given non-causal features and object locations. This serves as an indicator for spurious correlations existing in datasets. The proposed definition first provides a quantitative way for measuring the distributional shifts for OOD-OD to the best of our knowledge. We leave it for future work to compute the numerical values of shifts given an object detection dataset.

## A.2 PROOF

**Proposition A.2.** *For any probability functions $p$ and $q$ of training distribution and testing distribution, $D_{diversity}$ and $D_{correlation}$ are inclusively bounded between 0 and 1.*

---

[1]We assume no category label ($\mathbf{Y}_1$) shifts for simplicity and without loss of generality

Table 6: Illustration of the two settings on Sim2real. $\text{sim}_{\text{train}}$ and $\text{sim}_{\text{val}}$ indicate the training set and validating set from original Sim10K (Johnson-Roberson et al., 2016) while $\text{city}_{\text{train}}$ and $\text{city}_{\text{val}}$ are from original Cityscapes (Cordts et al., 2016). **Quantity** indicates the number of images. **Total** counts the total number of training and testing domains respectively.

| Setting | Split | Train | Test | Quantity | Total |
|---|---|---|---|---|---|
| part-sim-part-real | $\text{sim}_{\text{train}}$ | $\checkmark$ | | 8000 | 8500 |
| | $\text{sim}_{\text{val}}$ | | $\checkmark$ | 1000 | |
| | $\text{city}_{\text{train}}$ | | $\checkmark$ | 2975 | 3975 |
| | $\text{city}_{\text{val}}$ | $\checkmark$ | | 500 | |
| all-sim-all-real | $\text{sim}_{\text{train}}$ | $\checkmark$ | | 8000 | 9000 |
| | $\text{sim}_{\text{val}}$ | $\checkmark$ | | 1000 | |
| | $\text{city}_{\text{train}}$ | | $\checkmark$ | 2975 | 3475 |
| | $\text{city}_{\text{val}}$ | | $\checkmark$ | 500 | |

*Proof.* Obviously, both $D_{\text{diversity}}$ and $D_{\text{correlation}}$ are positive. Then, we prove the upper bound by the triangle inequality as followed:

$$D_{\text{diversity}} = \frac{1}{2}\int_{\mathbf{S}} \mid p(z) - q(z) \mid dz \leq \frac{1}{2}\int_{\mathbf{S}} [p(z) + q(z)]\, dz \leq 1 \tag{6}$$

Similarly, we have the following inequality:

$$\begin{aligned}
D_{\text{correlation}} &= \frac{1}{2}\int_{\mathbf{T}} \sqrt{p(z)\cdot q(z)} \sum_{y_1,y_2} \mid p(y_1, y_2|z) - q(y_1, y_2|z) \mid dz \\
&\leq \frac{1}{2}\int_{\mathbf{T}} \sqrt{p(z)\cdot q(z)} \sum_{y_1,y_2} [p(y_1, y_2|z) + q(y_1, y_2|z)]\, dz \\
&= \frac{1}{2}\int_{\mathbf{T}} 2\sqrt{p(z)\cdot q(z)}\, dz = \int_{\mathbf{T}} \sqrt{p(z)\cdot q(z)}\, dz \leq 1
\end{aligned} \tag{7}$$

The second inequality is due to triangle inequality. □

### A.3 IMPLEMENTATION DETAILS

To evaluate the object detection algorithms, we use the models and the pre-trained weights provided by mmdetection (Chen et al., 2019). For domain generalization algorithms on OOD generalization object detection, we derive the implementations using Faster R-CNN (Ren et al., 2015) with ResNet-50 FPN backbone (He et al., 2016) from torchvision. The whole network is optimized by Stochastic Gradient Descent with learning rate 0.02, momentum 0.9 and weight decay 0.0005.

### A.4 FURTHER RESULTS

**Task complexity.** To analyse the IID condition on CtrlShift, which indicates both $D_{\text{correlation}}$ and $D_{\text{diversity}}$ equal zero, we propose a hyper-parameter task complexity $\alpha$ to measure the difficulty of the task. The difficulty is adjusted by using $1 - \alpha$ percent novel data in the testing set in addition to the original training data. The experimental results are shown in Figure 7. The generalization ability of each algorithm drops with the increase of task complexity.

**Sim2real benchmark.** The training set of the Sim2real results reported in the main manuscript comprises the training data from Sim10K (Johnson-Roberson et al., 2016) and the validating data from Cityscapes (Cordts et al., 2016), while the testing set comprises the training data from Cityscapes (Cordts et al., 2016) and the validating data from Sim10K (Johnson-Roberson et al., 2016) (noted by part-sim-part-real, more details can be found in Table A.4). We reported the experimental results on all-sim-all-real in Table 7 and Table 8.

**Full results.** Table 9, 10, 11, 12 and 13 are the full experimental results of Table 3 and 4, including AP, $AP_{50}$, $AP_{75}$, $AP_s$, $AP_m$ and $AP_l$ evaluation metrics.

Table 7: The experimental results of object detection algorithms on the all-sim-all-real of Sim2real. Mem (GB)† and Inf time (fps)† are from mmdetection (Chen et al., 2019).

| Detector | Backbone | Mem† | fps† | AP |
|---|---|---|---|---|
| Faster R-CNN (Ren et al., 2015) | X-101 | 10.3 | 9.4 | 35.6 |
| RetinaNet (Lin et al., 2017b) | X-101 | 10.0 | 8.7 | 38.0 |
| Mask R-CNN (He et al., 2017) | X-101 | 10.7 | 8.0 | 36.7 |
| CornetNet (Law & Deng, 2018) | Hourglass104 | 13.9 | 4.2 | 21.6 |
| YOLOv3 (Redmon & Farhadi, 2018) | DarkNet-53 | 7.4 | 48.1 | 28.2 |
| FCOS (Tian et al., 2019) | X-101 | 10.0 | 9.7 | 37.9 |
| Cascade R-CNN (Cai & Vasconcelos, 2019) | X-101 | 10.7 | - | 40.5 |
| MS R-CNN (Huang et al., 2019) | R-X101 | 11.0 | 8.0 | 35.7 |
| Libra R-CNN (Pang et al., 2019) | X-101 | 10.8 | 8.5 | 35.3 |
| DH R-CNN (Wu et al., 2019) | R-50 | 6.8 | 9.5 | 33.8 |
| VarifocalNet (Zhang et al., 2020a) | X-101 | - | - | 42.3 |
| Sparse R-CNN (Sun et al., 2020) | R-101 | - | - | 40.3 |
| Deformable (Zhu et al., 2021) | R-50 | - | - | 37.4 |
| YOLOX (Ge et al., 2021) | YOLOX-x | 28.1 | - | 36.4 |

Table 8: The experimental results of domain generalization algorithms on the all-sim-all-real of Sim2real.

| Algorithm | hyper-parameters | AP |
|---|---|---|
| ERM (Vapnik, 1998) | - | 32.8 |
| IB-ERM (Ahuja et al., 2021) | $\lambda_{ib} = 100$ | 18.3 |
| IRM (Arjovsky et al., 2019) | $\lambda_{irm} = 1$ | 32.7 |
| MMD (Li et al., 2018b) | $\gamma_{mmd} = 1$ | 33.2 |
| CORAL (Sun & Saenko, 2016) | $\gamma_{mmd} = 1$ | 32.5 |
| VREx (Krueger et al., 2021) | $\lambda_{vrex} = 1$ | 32.4 |
| GS (Pezeshki et al., 2020) | $\lambda_{reg} = 0.1$ | 31.4 |
| IGA (Koyama & Yamaguchi, 2021) | $\lambda_{penalty} = 1000$ | 33.4 |
| GroupDRO (Sagawa et al., 2019) | $\eta_{groupdro} = 0.01$ | 31.9 |

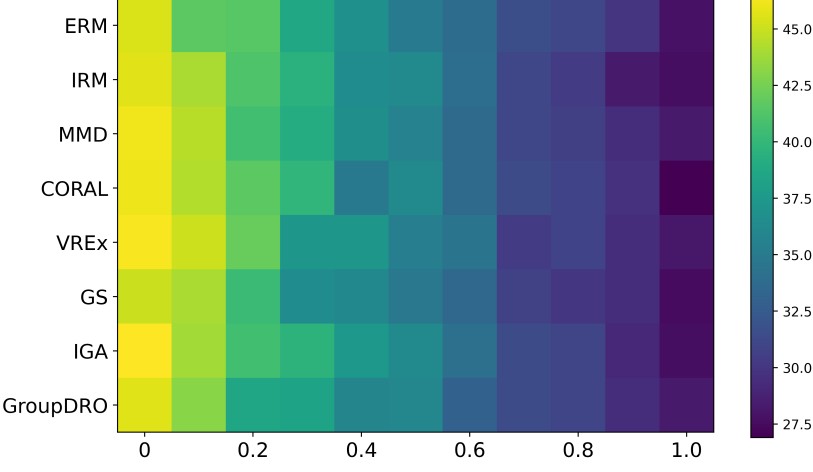

Figure 7: X-axis is task complexity $\alpha$. Each block indicates the AP(%).

Table 9: Experimental results of detectors on Sim2real.

| Detector | Backbone | AP | $AP_{50}$ | $AP_{75}$ | $AP_s$ | $AP_m$ | $AP_l$ |
|---|---|---|---|---|---|---|---|
| Faster R-CNN | X-101 | 52.1 | 73.6 | 55.2 | 22.3 | 58.6 | 82.3 |
| RetinaNet | X-101 | 49.3 | 72.2 | 51.0 | 16.8 | 56.2 | 81.6 |
| Mask R-CNN | X-101 | 52.9 | 74.2 | 56.4 | 23.2 | 60.0 | 82.7 |
| CornetNet | Hourglass104 | 27.8 | 40.2 | 28.7 | 6.6 | 34.4 | 46.7 |
| YOLOv3 | DarkNet-53 | 38.2 | 62.7 | 40.0 | 10.2 | 42.2 | 70.0 |
| FCOS | X-101 | 50.8 | 72.1 | 53.1 | 21.4 | 56.6 | 82.1 |
| Cascade R-CNN | X-101 | 53.1 | 74.1 | 56.2 | 23.2 | 59.2 | 83.6 |
| MS R-CNN | R-X101 | 52.5 | 73.7 | 56.0 | 23.0 | 59.5 | 82.3 |
| Libra R-CNN | X-101 | 51.4 | 72.1 | 55.0 | 22.0 | 59.7 | 81.5 |
| DH R-CNN | R-50 | 52.2 | 73.4 | 56.1 | 23.6 | 59.2 | 81.2 |
| VarifocalNet | X-101 | 56.1 | 75.3 | 59.4 | 25.9 | 63.9 | 85.9 |
| Sparse R-CNN | R-101 | 54.1 | 76.8 | 57.7 | 26.7 | 59.3 | 81.9 |
| DETR | R-50 | 37.8 | 63.9 | 38.3 | 10.2 | 38.0 | 72.1 |
| Deformable | R-50 | 53.3 | 78.0 | 57.0 | 25.6 | 59.7 | 80.2 |
| Swin Transformer | Swin-B | 58.4 | 78.5 | 63.5 | 31.4 | 65.5 | 84.2 |
| YOLOX | YOLOX-x | 51.1 | 70.3 | 53.7 | 19.3 | 58.4 | 84.4 |

Table 10: Experimental results of detectors on Weather.

| Detector | Backbone | AP | $AP_{50}$ | $AP_{75}$ | $AP_s$ | $AP_m$ | $AP_l$ |
|---|---|---|---|---|---|---|---|
| Faster R-CNN | X-101 | 35.3 | 56.1 | 36.3 | 14.5 | 37.5 | 49.5 |
| RetinaNet | X-101 | 34.7 | 54.3 | 35.8 | 12.8 | 36.9 | 49.4 |
| Mask R-CNN | X-101 | 34.8 | 55.2 | 35.8 | 14.9 | 36.9 | 47.9 |
| CornetNet | Hourglass104 | 29.7 | 46.1 | 30.6 | 12.2 | 34.3 | 38.4 |
| YOLOv3 | DarkNet-53 | 27.0 | 47.9 | 26.1 | 8.7 | 29.4 | 40.5 |
| FCOS | X-101 | 35.6 | 55.7 | 36.4 | 14.8 | 37.9 | 48.8 |
| Cascade R-CNN | X-101 | 36.3 | 56.4 | 37.8 | 15.1 | 38.2 | 51.7 |
| MS R-CNN | R-X101 | 35.5 | 56.1 | 36.6 | 14.6 | 37.4 | 50.2 |
| Libra R-CNN | X-101 | 34.9 | 54.8 | 36.3 | 14.0 | 37.4 | 49.5 |
| DH R-CNN | R-50 | 35.3 | 55.7 | 36.5 | 14.5 | 37.2 | 50.5 |
| VarifocalNet | X-101 | 38.7 | 59.3 | 39.8 | 16.0 | 41.0 | 54.1 |
| Sparse R-CNN | R-101 | 36.3 | 57.5 | 37.1 | 15.6 | 38.5 | 50.8 |
| DETR | R-50 | 23.5 | 41.7 | 22.5 | 4.9 | 23.2 | 40.1 |
| Deformable | R-50 | 35.1 | 56.7 | 35.3 | 14.7 | 37.2 | 49.8 |
| Swin Transformer | Swin-B | 36.6 | 56.2 | 38.2 | 15.8 | 39.7 | 51.1 |
| YOLOX | YOLOX-x | 33.1 | 51.7 | 34.1 | 11.5 | 35.4 | 47.6 |

## A.5 FURTHER DISCUSSION

High accuracy in IID may be the results of over-fitting since the spurious correlation exists in both the training and the testing distribution. Recently proposed methods, such as Deformable DETR, Sparse R-CNN and Swin Transformer, significantly outperform classic Faster R-CNN on IID dataset, however, have similar performance on OOD datasets. We can conclude that these methods make prediction based on spurious correlated representations existed in IID data and not existed in OOD data.

Defined by our theory in Appendix A.1, diversity shift is embodied by the novel features not shared by environments while correlation shift is caused by the shared non-causal features. Experiment results in Section 5.2 show that VREx and IRM obtain better OOD generalization ability against correlation shift than diversity shift, and it is a challenge to tackle diversity shift.

Current study starts to explore the impact of model architectures on OOD performance. NAS-OoD (?) applies a neural architecture search strategy to find the architecture with optimal OOD generalization ability and the results demonstrate that model architecture can significantly influence OOD accuracy. Some detectors have relatively consistent performance on OOD, such as VarifocalNet,

Table 11: Experimental results of detectors on Scene.

| Detector | Backbone | AP | $AP_{50}$ | $AP_{75}$ | $AP_s$ | $AP_m$ | $AP_l$ |
|---|---|---|---|---|---|---|---|
| Faster R-CNN | X-101 | 36.0 | 56.3 | 38.0 | 15.9 | 38.8 | 54.7 |
| RetinaNet | X-101 | 35.8 | 56.1 | 37.4 | 13.5 | 39.5 | 55.9 |
| Mask R-CNN | X-101 | 35.8 | 56.2 | 37.5 | 15.7 | 38.2 | 54.3 |
| CornetNet | Hourglass104 | 30.8 | 46.1 | 32.6 | 13.8 | 32.7 | 45.9 |
| YOLOv3 | DarkNet-53 | 28.2 | 49.7 | 27.1 | 10.0 | 32.3 | 44.2 |
| FCOS | X-101 | 36.5 | 56.7 | 38.7 | 15.9 | 39.1 | 54.8 |
| Cascade R-CNN | X-101 | 36.7 | 56.7 | 38.1 | 15.0 | 39.3 | 58.9 |
| MS R-CNN | R-X101 | 35.7 | 56.1 | 37.2 | 15.3 | 38.4 | 54.4 |
| Libra R-CNN | X-101 | 35.5 | 55.6 | 36.8 | 15.9 | 38.3 | 54.9 |
| DH R-CNN | R-50 | 35.2 | 55.3 | 36.9 | 15.9 | 38.2 | 54.7 |
| VarifocalNet | X-101 | 39.0 | 58.8 | 40.8 | 18.1 | 41.4 | 59.4 |
| Sparse R-CNN | R-101 | 36.4 | 57.4 | 37.4 | 18.4 | 39.1 | 56.0 |
| DETR | R-50 | 24.4 | 42.8 | 23.8 | 6.3 | 25.3 | 47.4 |
| Deformable | R-50 | 35.3 | 55.8 | 36.2 | 15.3 | 37.9 | 54.6 |
| Swin Transformer | Swin-B | 32.0 | 50.1 | 33.7 | 14.7 | 36.7 | 48.3 |
| YOLOX | YOLOX-x | 34.8 | 54.1 | 36.2 | 13.4 | 37.7 | 53.4 |

Table 12: Experimental results of detectors on Time.

| Detector | Backbone | AP | $AP_{50}$ | $AP_{75}$ | $AP_s$ | $AP_m$ | $AP_l$ |
|---|---|---|---|---|---|---|---|
| Faster R-CNN | X-101 | 26.4 | 45.2 | 26.3 | 9.7 | 25.2 | 39.7 |
| RetinaNet | X-101 | 24.6 | 43.1 | 24.3 | 7.8 | 23.5 | 37.8 |
| Mask R-CNN | X-101 | 26.8 | 45.8 | 26.5 | 9.5 | 25.8 | 39.8 |
| CornetNet | Hourglass104 | 22.5 | 37.8 | 22.3 | 8.6 | 23.4 | 29.6 |
| YOLOv3 | DarkNet-53 | 19.2 | 34.5 | 18.7 | 5.1 | 18.3 | 31.3 |
| FCOS | X-101 | 24.8 | 42.5 | 24.4 | 9.2 | 23.6 | 36.2 |
| Cascade R-CNN | X-101 | 24.1 | 41.0 | 23.6 | 8.7 | 21.9 | 37.6 |
| MS R-CNN | R-X101 | 25.5 | 43.4 | 25.3 | 9.2 | 23.8 | 39.0 |
| Libra R-CNN | X-101 | 25.5 | 43.7 | 25.2 | 8.9 | 24.6 | 39.2 |
| DH R-CNN | R-50 | 25.5 | 43.9 | 25.4 | 9.4 | 24.9 | 37.6 |
| VarifocalNet | X-101 | 27.4 | 45.6 | 27.0 | 10.4 | 26.0 | 39.3 |
| Sparse R-CNN | R-101 | 28.2 | 48.3 | 27.7 | 10.6 | 27.2 | 41.4 |
| DETR | R-50 | 15.8 | 30.9 | 13.7 | 3.2 | 14.7 | 28.2 |
| Deformable | R-50 | 26.1 | 46.2 | 25.1 | 10.1 | 25.4 | 39.0 |
| Swin Transformer | Swin-B | 28.9 | 47.6 | 29.4 | 10.5 | 28.6 | 42.3 |
| YOLOX | YOLOX-x | 29.4 | 48.0 | 29.2 | 9.4 | 28.2 | 44.0 |

Table 13: Generalization performance of detectors with OOD algorithms.

| Algorithm | hyper-parameters | Dataset | AP | AP$_{50}$ | AP$_{75}$ | AP$_s$ | AP$_m$ | AP$_l$ |
|---|---|---|---|---|---|---|---|---|
| ERM | - | | 44.1 | 64.0 | 47.1 | 13.8 | 41.2 | 68.4 |
| IB-ERM | $\lambda_{ib} = 0.1$ | | 42.5 | 64.4 | 44.9 | 12.6 | 39.2 | 67.0 |
| IRM | $\lambda_{irm} = 0.1$ | | 42.3 | 64.9 | 45.0 | 12.7 | 39.4 | 66.1 |
| MMD | $\gamma_{mmd} = 1$ | | 42.9 | 64.8 | 45.4 | 13.2 | 39.8 | 67.2 |
| CORAL | $\gamma_{mmd} = 10$ | Sim2real | 42.6 | 64.6 | 45.3 | 12.8 | 39.7 | 66.8 |
| VREx | $\lambda_{vrex} = 1$ | | 43.1 | 65.4 | 45.4 | 13.0 | 40.0 | 67.4 |
| GS | $\lambda_{reg} = 10$ | | 40.3 | 64.2 | 42.5 | 11.5 | 37.6 | 65.3 |
| IGA | $\lambda_{penalty} = 1$ | | 42.9 | 64.7 | 45.6 | 13.7 | 40.0 | 66.6 |
| GroupDRO | $\eta_{groupdro} = 1$ | | 42.8 | 65.0 | 45.4 | 12.8 | 39.9 | 67.1 |
| RSC | $\lambda_{rsc} = 10$ | | 41.0 | 63.6 | 43.2 | 10.8 | 38.6 | 64.9 |
| ERM | - | | 24.4 | 47.8 | 21.8 | 9.5 | 29.3 | 41.9 |
| IB-ERM | $\lambda_{ib} = 0.1$ | | 24.7 | 47.7 | 21.9 | 9.7 | 29.6 | 41.8 |
| IRM | $\lambda_{irm} = 0.1$ | | 24.9 | 48.3 | 22.1 | 9.7 | 29.9 | 41.9 |
| MMD | $\gamma_{mmd} = 1$ | | 24.1 | 47.2 | 21.5 | 9.4 | 29.0 | 41.6 |
| CORAL | $\gamma_{mmd} = 10$ | Weather | 24.4 | 47.3 | 21.9 | 9.7 | 29.2 | 41.6 |
| VREx | $\lambda_{vrex} = 1$ | | 24.7 | 48.4 | 21.8 | 9.6 | 29.6 | 41.4 |
| GS | $\lambda_{reg} = 10$ | | 19.9 | 39.2 | 17.5 | 8.0 | 24.6 | 35.4 |
| IGA | $\lambda_{penalty} = 1$ | | 24.6 | 47.9 | 21.7 | 9.9 | 29.6 | 41.2 |
| GroupDRO | $\eta_{groupdro} = 1$ | | 24.5 | 47.8 | 21.6 | 9.6 | 29.2 | 41.3 |
| RSC | $\lambda_{rsc} = 10$ | | 8.7 | 17.6 | 7.5 | 3.3 | 11.1 | 17.9 |
| ERM | - | | 24.1 | 46.7 | 21.2 | 9.7 | 29.4 | 46.5 |
| IB-ERM | $\lambda_{ib} = 0.1$ | | 23.4 | 45.6 | 20.4 | 9.5 | 28.6 | 45.7 |
| IRM | $\lambda_{irm} = 0.1$ | | 23.2 | 45.8 | 20.3 | 9.4 | 28.4 | 45.0 |
| MMD | $\gamma_{mmd} = 1$ | | 23.9 | 46.4 | 21.2 | 9.4 | 29.2 | 47.5 |
| CORAL | $\gamma_{mmd} = 10$ | Scene | 22.8 | 44.5 | 20.1 | 9.4 | 27.8 | 45.4 |
| VREx | $\lambda_{vrex} = 1$ | | 24.3 | 47.0 | 21.6 | 9.7 | 29.7 | 46.6 |
| GS | $\lambda_{reg} = 10$ | | 18.4 | 35.9 | 16.4 | 8.0 | 23.7 | 38.0 |
| IGA | $\lambda_{penalty} = 1$ | | 24.3 | 47.6 | 21.0 | 10.2 | 29.3 | 45.9 |
| GroupDRO | $\eta_{groupdro} = 1$ | | 23.6 | 46.1 | 21.1 | 10.1 | 29.0 | 45.7 |
| RSC | $\lambda_{rsc} = 10$ | | 8.8 | 17.9 | 7.4 | 3.5 | 11.6 | 21.5 |
| ERM | - | | 17.7 | 37.5 | 14.6 | 6.6 | 19.5 | 32.0 |
| IB-ERM | $\lambda_{ib} = 0.1$ | | 18.2 | 38.3 | 15.2 | 6.8 | 20.1 | 32.1 |
| IRM | $\lambda_{irm} = 0.1$ | | 17.5 | 37.5 | 14.2 | 6.6 | 19.3 | 30.6 |
| MMD | $\gamma_{mmd} = 1$ | | 18.4 | 38.4 | 15.4 | 7.0 | 20.2 | 31.4 |
| CORAL | $\gamma_{mmd} = 10$ | Time | 17.7 | 37.6 | 14.6 | 6.4 | 19.5 | 30.7 |
| VREx | $\lambda_{vrex} = 1$ | | 18.0 | 38.3 | 14.9 | 6.8 | 19.9 | 31.2 |
| GS | $\lambda_{reg} = 10$ | | 15.2 | 32.4 | 12.5 | 5.6 | 17.4 | 27.7 |
| IGA | $\lambda_{penalty} = 1$ | | 18.6 | 39.1 | 15.4 | 7.2 | 20.4 | 32.0 |
| GroupDRO | $\eta_{groupdro} = 1$ | | 18.1 | 38.1 | 15.1 | 6.9 | 20.1 | 31.4 |
| RSC | $\lambda_{rsc} = 10$ | | 7.0 | 15.0 | 5.7 | 2.5 | 8.2 | 14.7 |

the proposed varifocal loss (Zhang et al., 2020a) can be considered to improve OOD generalization ability.

