# OpenReview forum: "DetectBench: An Object Detection Benchmark for OOD Generalization Algorithms"
_ICLR.cc/2023/Conference — Submitted to ICLR 2023_

### Official Review · Reviewer_aJ6J · 2022-10-19

**Confidence:** 4
**Correctness:** 3
**Technical Novelty And Significance:** 2
**Empirical Novelty And Significance:** 2
**Recommendation:** 3

**Clarity, Quality, Novelty And Reproducibility:**

See details above, and the authors provide the code but I haven't run it for reproducibility.

**Strength And Weaknesses:**

Strong points:

1. The paper is well written.
2. Details of the datasets are provided.
3. The authors run the benchmark on both two kinds of distribution shifts, proposed by [1].
4. The authors introduce a new synthetic dataset CtrlShift, of which API can be modified to simulate the two kinds of shifts mentioned above.
5. This would be a good contribution to the community.
6. The authors extensively benchmark and run experiments on a wide range of detection algorithms.
7. Some algorithms from OoD image classification are also adapted to DetectBench.

Nevertheless, there are some concerns about the paper that make the findings not convincing. I am happy to raise the score if the authors can address the concerns.

Weak points:

1. The train-test split is still a bit weak. For example, in DomainBed [2], for PACS dataset they divide it into four domains. Using the training-domain splitting strategy, they train the models on three selected domains and test on the other. This is to determine if the distribution shift is actually present, or it is because a model or architecture simply performs better on a specific domain. In this paper, the authors split the Weather in BDD100K into six domains, but only select two domains to train and the rest four domains to test. What would happen if we simply flip the train-test split? Will the ranking still hold?
2. Missing the potential impact of the small-scale dataset on big models. Due to the dataset size, Sim2real has fewer data compared to BDD100K. This could make DETR models underperform compared to Faster R-CNN, as transformer models are generally data-hungry due to their size [5].
3. Missing justification for choosing a strong backbone for Faster R-CNN and other detectors, but a regular backbone for DETR. In the paper, the authors choose ResNeXt-101-64x4d for Faster R-CNN and other models but use ResNet50 for DETR and Deformable DETR. Backbone network plays a vital role in the performance, which has been observed in both IID Object Detection [3] and OoD classification [4]. I think it is better to use the same backbone in this table so that we can measure the effect of different detectors.
4. Missing results in (strong) Diversity Shift datasets. Diversity shift datasets such as Watercolor2k, Comic1k, Clipart1k [7] are missing from the paper. In the classification benchmark, the diversity shift datasets (PACS, VLCS, etc.) are much harder to imporve than correlation shift datasets (NICO, CelebA). From my experience, all OoD classification algorithms achieve very high accuracy by simply using ResNet50 as backbone on Correlation shift datasets, making gains of algorithms on this shift extremely minor.
5. The findings are limited in novelty and not very convincing. Section 5.1 highlights some findings in the paper, but I do not find two points very “surprising” nor convincing:
5.1) Finding 1 (The enormous achievements of object detection on IID datasets are marginal on the OOD condition): It is widely known that IID is a strong assumption in real-world use cases, and most models are sensitive to OoD tasks. This is not limited to only image classification, but also object detection, which has already been observed in BDD100K paper [6].
5.2) Finding 3 (The generalization inconsistency between classification and object detection of domain generalization algorithms happens among different detectors): As pointed out above in Weak points 2 and 3, Sim2real is a relatively small dataset for DETR (8.5k images), which possibly means results in Table 5 are not reflecting an accurate performance of DETR. Based on Table 3, the reviewer thinks Deformable DETR is a more suitable choice, and the conclusion “the degeneration of domain generalization algorithms has little relevance to the detectors” is probably a bit overclaimed.

Minor fixes:
1. Figure 4 X-axis is a bit hard to read. The authors may consider using vertical labels.
2. Finding 3 in Section 5.1, “Table 4.2” should be “Table 5”

References

[1] Ye, Nanyang, et al. “Ood-bench: Benchmarking and understanding out-of-distribution generalization datasets and algorithms.” In CVPR (2021).

[2] Gulrajani, Ishaan, and Lopez-Paz, David. “In search of lost domain generalization.” In ICLR (2020).

[3] Zhu, Xizhou, et al. “Deformable DETR: Deformable transformers for end-to-end object detection.” In ICLR (2021).

[4] Angarano, Simone, et al. “Back-to-bones: Rediscovering the role of backbones in domain generalization.” arXiv preprint arXiv:2209.01121 (2022).

[5] Wang, Wen, et al. “Towards Data-Efficient Detection Transformers.” In ECCV (2022).

[6] Yu, Fisher, et al. “Bdd100k: A diverse driving video database with scalable annotation tooling.” arXiv preprint arXiv:1805.04687 2.5 (2018): 6.

[7] Inoue, Naoto, et al. “Cross-domain weakly-supervised object detection through progressive domain adaptation.” In CVPR (2018).

**Summary Of The Paper:**

This paper proposes DetectBench - a benchmark for Out-of-Distribution (OoD) Object Detection, as most works mainly focus on OoD image classification. After introducing a new train-test split on existing datasets and newly proposed dataset CtrlShift, the paper benchmarks a wide range of detectors (Faster R-CNN, DETR, etc.) with different backbones, self-implemented OoD image classification algorithms (ERM, RSC, GroupDRO, etc.), and different detectors with same backbone (Faster R-CNN, RetinaNet, DETR) on these datasets. The main findings are that IID object detection algorithms are marginal on OoD object detection, and the authors propose a new synthetic dataset, which can be used to simulate the two types of distribution shifts based on user’s inputs.

Object detection is an important task for many applications, including safety-critical ones such as self-driving cars and health care. Accordingly, improving the ability of object detectors to deal with unseen distribution shifts is highly relevant. This benchmark is thus well-motivated.

**Summary Of The Review:**

I agree the proposed benchmark is well-motivated and the task is important for the research community. However, with the findings above, I currently give the paper a borderline reject score.

Update: My concerns are not entirely addressed after the rebuttal. I will keep the rejection score for this submission, due to unconvincing experimental results (without cross-validation), weak diversity shift benchmark, the limited novelty of the findings, missing related works, and the proposed benchmarks overlapping with previous works.

---

> ### Author Response · Authors · 2022-11-14
> **2.Response to Reviewer aJ6J**
>
> **C1: The train-test split is still a bit weak.**
>
> The purpose of the train-test split is to model the real scenarios cases where autonomous vehicles have to detect objects correctly even under untrained environments, which is a key ability for safe self-driving vehicles. As object detection datasets are usually much larger than image classification datasets, we did not do the cross-fold validation due to the limitation of computational resources. We flip the train-test split of OOD-Weather and the results reported below show that the flip will not affect the result that existing OOD generalization algorithms are hard to work for object detection tasks. The results should be similar under different splits. More split results can be added upon request.
>
> |Algorithm|Detector|AP|AP50|AP75|AP(s)|AP(m)|AP(l)|
> | :----- | :----:| :----:| :----: | :----: | :----: | :----: | :----:|
> |ERM|Faster R-CNN + R-50|22.1|44.5|19.0|9.2|27.2|39.6|
> |IRM|Faster R-CNN + R-50|22.1|44.4|19.2|9.0|26.7|40.3|
> |VREx|Faster R-CNN + R-50|21.9|43.9|19.0|8.8|26.6|39.6|
> |MMD|Faster R-CNN + R-50|22.1|44.3|19.2|9.3|27.0|40.0|
>
> **C2: Missing the potential impact of the small-scale dataset on big models.**
>
> OOD generalization algorithms should have higher data efficiency as they aim to generalize to untrained data distributions. As human beings, we do not need millions of data to detect objects correctly against their superficial changes. Recent autonomous vehicle accidents alert us that we cannot always rely on large-scale data for safety. It is not possible or hard to collect all cases to avoid mistakes. Without OOD generalization abilities, practical scenarios, such as medical systems [1] and autonomous vehicles [2] would fail unexpectedly in unseen environments which are exponentially hard to collect.
>
>
> **C3: Missing justification for choosing a strong backbone for Faster R-CNN and other detectors.**
>
> The main claim of our benchmark is the performance inconsistency of the latest object detection works on IID and OOD conditions and the failure of OOD generalization algorithms on improving them for OOD conditions. As shown in Table 3 in the revised version, DETR + ResNet-50 backbone has comparable performance with Faster RCNN + ResNeXt-101 on MS COCO, however, DETR + ResNet-50 drops significantly on our OOD benchmarks. Moreover, we study DETR + ResNet-101 on OOD-Weather and the results further demonstrate our conclusion.
>
> |Detector|Backbone|Env|AP|AP50|AP75|AP(s)|AP(m)|AP(l)|
> | :----- | :----:| :----: | :----: | :----: | :----: | :----:| :----: | :----:|
> |Faster R-CNN| ResNeXt-101 | MS COCO |42.1|63.0|46.3|24.8|46.2|55.3|
> |DETR| ResNet-50 | MS COCO |42.0|62.4|44.3|21.0|45.8|61.0|
> |DETR| ResNet-101 |MS COCO|43.5|63.8|46.3|21.9|47.9|61.9|
> |Faster R-CNN| ResNeXt-101 | Weather |35.3 | 56.1 | 36.3 | 14.5 | 37.5 | 49.5|
> |DETR| ResNet-50 | Weather|23.5|41.7|22.5|4.9|23.2|40.1|
> |DETR| ResNet-101 |Weather|25.0|44.2|23.9|5.8|25.1|40.4|
>
> **C4: Missing results in (strong) Diversity Shift datasets..**
>
> According to our theoretical definition of diversity shift, the Sim2Real, Weather, Scene, Time datasets are dominated by diversity shift. For example, similar to PACS dataset where each domain is of a different image style, the training and test set's image styles are largely different in Sim2Real dataset ((Lower resolution computer graphics generated images vs. photographs).
>
> **C5: The findings are limited.**
>
> Recent OOD generalization algorithms demonstrate huge performance improvements over ERM on image classification tasks. While most algorithms claim to be general, we found that the improvements are hardly seen in OOD object detection tasks. In this paper, we want to raise concerns in the OOD generalization research community to develop practically useful algorithms beyond toy image classification tasks, such as object detection. Improvements in robust object detection methods capable of OOD generalization are key for safer autonomous driving. This may save lives and we hope more researchers could pay attention to this instead of just showing improvements on toy datasets and upgrading SOTAs.
>
> [1] Causality matters in medical imaging
>
> [2] Dark model adaptation: Semantic image segmentation from daytime to nighttime

---

> > ### Comment · Reviewer_aJ6J · 2022-11-22
> > **rebuttal update**
> >
> > C1: According to me, cross-validation should be adapted to the standard experimental protocol to prevent new approaches from overfitting the fixed split.
> >
> > C2: I recommend adding assumptions or evidence to help explain why existing network architectures and DG methods fail to generalize on proposed datasets. Many factors could contribute to the author’s reported results. It’s possible some network architectures and DG methods are not well explored in this paper. (Reviewer Sn2d also mention this point)
> >
> > C3: Thanks for the new results. I recommend using the same backbone for a fair comparison.
> >
> > C4: I didn’t find any clear evidence that the proposed datasets show strong diversity shift datasets. Please refer to [1] for more details on the diversity shift.
> >
> > C5: I still have the feeling the findings are limited in novelty and not very convincing after reading the response. Please see my details comments above.
> >
> > After reading other comments, I find this work overlaps with previous works. Please carefully list the difference with previous works in the related work.
> > More related work:[8]
> >
> > [8] Benchmarking Robustness in Object Detection: Autonomous Driving when Winter is Coming

---

> > > ### Author Response · Authors · 2022-11-22
> > > **3.Response to Reviewer aJ6J**
> > >
> > > Thanks for your response.
> > >
> > > **C1:**
> > > It is easy to do cross-validation with different train-test splits using our codes. Besides, the benchmark results on three train-test splits have already demonstrated the performance inconsistency of OOD algorithms and the further results provided in C1 in 2.Response to Reviewer aJ6J, also suggest this conclusion. Due to the time limitation, we can provide results upon request.
> > >
> > > **C2:**
> > > Thanks for your insightful suggestions. The main purpose of the paper is to raise concerns in OOD generalization research community about the applicability of the algorithms beyond classification. We hope our work can serve as a foothold for future OOD object detection research.
> > >
> > > **C3:**
> > > The following table lists the results of ResNet-101 backbone.
> > >
> > > |Detector|Backbone|Env|AP|AP50|AP75|AP(s)|AP(m)|AP(l)|
> > > | :----- | :----:| :----: | :----: | :----: | :----: | :----:| :----: | :----:|
> > > |Faster R-CNN| ResNet-101 |MS COCO |39.4| 60.1| 43.1| 22.4| 43.7|51.1|
> > > |DETR| ResNet-101 |MS COCO|43.5|63.8|46.3|21.9|47.9|61.9|
> > > |Faster R-CNN| ResNet-101 | Weather |34.2 | 55.1 | 34.9 | 14.2 | 36.4 | 48.4 |
> > > |DETR| ResNet-101 |Weather|25.0|44.2|23.9|5.8|25.1|40.4|
> > >
> > > **C4:**
> > > According to OoD-Bench [1], diversity shift is embodied by the features set $S := \{ z \in Z_2 | p(z) \cdot q(z) = 0 \}$, where $Z_2$ is the non-causal features. $p(\cdot)$ and $q(\cdot)$ indicate the training distribution and testing distribution, respectively. Consider the following example: "rain" is one of the non-causal information, thus, the features $S$ extracted from "rain" belongs to $Z_2$. However, $S$ exists in the training data of OOD-Weather and not exists in the testing data, and this is the diversity shift. This also happens in the other three OOD environments.
> > >
> > > **C5:**
> > > We would like to quote a paragraph in our summarization to re-emphasize the impact of this work on the OOD research community. "In this work, we propose an OOD object detection benchmark for OOD algorithms (domain generalization algorithms that were mainly evaluated on (toy) image classification tasks. It is found that the huge improvements of recent OOD generalization algorithms reported on image classification tasks fail to transfer to OOD object detection tasks, though many claimed to be a general algorithm. Besides, recent progress on IID object detection tasks is not so prominent on OOD object detection tasks. In this paper, we hope to draw researchers’ attention to make a step forward in marrying important topics of OOD generalization and other practical tasks, such as object detection together, since this is one of the keys to safer autonomous driving which could be life-saving."
> > >
> > > **C6:**
> > > Difference between our work and [8]
> > >
> > > 1. According to Appendix C in [8], the training set based on BDD100K comprises all clear weather images from the origin training set and the validation set comprises clear, rainy and snowy conditions. While we strictly control the training and testing domains do not overlap.
> > > 2. The image corruption in [8] is considered as noise to validate models' robustness against different corruptions and the train-test set is still IID. Moreover, according to [1], distribution shifts are caused by non-causal features, such as "weather", and the noises are not in the features scopes.
> > > 3. We are the first to construct multi-domain for the training set and the testing set, which is widely acknowledged by previous OOD algorithms [9-12] that the diversity of the training distribution is one of the keys to achieving OOD generalization.
> > >
> > > Due to the manuscript revision deadline, we will add a section to illustrate our train-test split in the later version.
> > >
> > >
> > > [9] Invariant Risk Minimization
> > >
> > > [10] Distributionally Robust Neural Networks for Group Shifts: On the Importance of Regularization for Worst-Case Generalization
> > >
> > > [11] Domain Generalization with Adversarial Feature Learning
> > >
> > > [12] Deep CORAL: Correlation Alignment for Deep Domain Adaptation

---

> ### Author Response · Authors · 2022-11-14
> **1.Response to Reviewer aJ6J**
>
> Thank you very much for your constructive comments. The main point of this paper is to raise concerns in OOD generalization research community to systematically investigate OOD generalization algorithms that can generalize to other tasks, such as object detection, beyond image classification. Certainly, the issue of the performance degradation of object detection algorithms against different domains caused by different sensors, sim2real, and etc has been empirically found by previous papers and remains largely unsolved. While recent advancements in OOD generalization algorithms have achieved significant performance improvement, we first systematically investigate whether these improvements are transferable to object detection tasks, which is key to safer autonomous driving. We hope this can serve as initial steps for marrying the fields of (largely theoretical) OOD generalization and object detection and a benchmark for further OOD generalization algorithms that aims not only at standard image classification OOD generalization tasks but also at object detection tasks that are widely used in practical computer vision.

---

### Official Review · Reviewer_Sn2d · 2022-10-21

**Confidence:** 4
**Correctness:** 2
**Technical Novelty And Significance:** 2
**Empirical Novelty And Significance:** 2
**Recommendation:** 3

**Clarity, Quality, Novelty And Reproducibility:**

The paper requires major revision on writing and structuring. Especially, I find it is very hard to have a clear picture of the experiment setup of the work that greatly compromises the quality of the work.

For instance, in Table 3 and 4, there should include ID performance as the baseline. However, it is either very difficult to find or generally unclear what is the ID training set for each domain shift.

In Table 5, if the first line is the vanilla version of object detection training, it is unclear why applying OOD techniques would result in performance degradation.

In 5.2, the experiment and visualization are not clear and sufficient to support the conclusion that correlation shift is easier than diversity shift to tackle.



**Strength And Weaknesses:**

Strength:
OOD generalization is an important task and is underexplored for object detection algorithms. This paper targets at this important problem. The authors proposed a OOD benchmark for object detection and compared a number of object detectors, empirically showing that existing solutions still struggle at dealing with domain shifts.

Weakness:
1. DetectBench does not differ much from previous benchmarks in this area. For example, [1] uses the eight shared categories in Cityscapes, Foggy Cityscape and BDD100k, and the Car category on these three datasets and additionally SIM 10k and KITTI. [2] also uses BDD100K, Cityscapes, Sim10K and KITTI as four domains, and train on three of them and test on the last one. The proposed CtrlShift dataset is different to these benchmarks, but is also limited to only two-dimension shift, and is not a very comprehensive new dataset with a large syn2real gap.

2. While considering a good number of object detectors, the authors mainly evaluate their performance on the constructed benchmarks. Beyond the general observation that these methods are not robust under domain shifts, there is no further insightful discussion, e.g., correlation between ID and OOD performance, different generalization behaviors under different shifts, how the model architecture would affect ID and OOD performance. More importantly, there generally miss a good justification that DetectBench is a qualified benchmark that allows cross-compare different methods. Especially, it is already shown by many previous works that current object detectors fail at generalizing to OOD scenes [1] [2] [3].

3. The compared OOD generalization baselines are not adapted to the task of object detection. All the OOD generalization methods are only used for the classification branch of the object detectors. The localization branch is not considered. It is important that this new benchmark should also investigate these OOD generalization techniques developed for object detection.

[1] Domain-Invariant Disentangled Network for Generalizable Object Detection
[2] Towards Domain Generalization in Object Detection
[3] OOD-CV: A Benchmark for Robustness to Out-of-Distribution Shifts of Individual Nuisances in Natural Images

**Summary Of The Paper:**

Building on top of BDD100k, Cityscapes and Sim10k, the proposed DetectBench in this work is an object detection benchmark for evaluating the OOD generalization performance of object detectors. The benchmark consists of 4 types of domain shifts, i.e., sim2real, weather, scene and time. The authors compared a number of object detectors on DetechBench, showing their yet poor OOD generalization performance. Using OOD generalization techniques developed for classification tasks, the object detection performance remains unsatisfactory, indicating OOD generalization of object detection is an underexplored task.

**Summary Of The Review:**

This work targets an important problem, but the usefulness, novelty and effectiveness of the proposed benchmark are not convincing.

---

> ### Author Response · Authors · 2022-11-14
> **2.Response to Reviewer Sn2d**
>
> **4: Experiment setup.**
>
> For our four benchmarks, we first construct the non-overlap training domain and testing domain to create OOD distribution shifts and all models only have the access to the training distribution. Then we train our baseline methods on the training domain data and evaluate them on the testing domain data. More details for model selection can be found in 4.3.
>
> **5: There should include ID performance as the baseline in Table 3 and 4.**
>
> The BDD100K, Sim10K and Cityscapes datasets are split into different domains to construct OOD setting and it is hard to build the IID train-test sets based on these domains. We provide the results on MS COCO dataset as the IID baselines in Table 3 in the revised version. Since the OOD generalization is harder, the IID performances are generally higher than OOD performances for the same method. We also found that the IID performances may not be a good indicator of OOD performances. For example, the classic l Faster-RCNN achieves similar OOD performances as recently proposed Cascade R-CNN and Deformable DETR, but Cascade R-CNN and Deformable DETR achieve much higher accuracies on IID dataset.
>
> **6: It is unclear why applying OOD techniques would result in performance degradation..**
>
> The degeneration of OOD techniques is reasonable. As the OOD generalization algorithms are mostly designed for classification, transferring them to object detection may violate some assumptions. An example is Mixup [5], objects are located in the center of the image in classification while objects are dispersed on the whole image, thus, the image-level mixup will mix the objects with backgrounds which does not make sense.
>
> **7: The experiment and visualization are not clear and sufficient to support the conclusion that correlation shift is easier than diversity shift to tackle.**
>
> From the visualization, we can conclude that IRM and VREx may help mitigate performance degradation when confronted with correlation shifts, and there is little evidence that supports that the correlation shift is easier than diversity shift to tackle in object detection.
>
> [1] Domain-Invariant Disentangled Network for Generalizable Object Detection
>
> [2] Towards Domain Generalization in Object Detection
>
> [3] OOD-CV: A Benchmark for Robustness to Out-of-Distribution Shifts of Individual Nuisances in Natural Images
>
> [4] OoD-Bench: Quantifying and Understanding Two Dimensions of Out-of-Distribution Generalization
>
> [5] Improve Unsupervised Domain Adaptation with Mixup Training
>
> [6] NAS-OoD: Neural Architecture Search for Out-of-Distribution Generalization
>
> [7] VarifocalNet: An IoU-aware Dense Object Detector

---

> ### Author Response · Authors · 2022-11-14
> **1.Response to Reviewer Sn2d**
>
> **1-1: DetectBench does not differ much from previous benchmarks.**
>
> Compared with [1][2], previous works use four datasets as four domains, the algorithms are trained on three domains and tested on the other domain. However, this evaluation protocol is in stark contrast with ours. Because the environmental information is largely overlapped in these four datasets. Other OOD factors, such as time of the day, weather, and seasons can be largely ignored. For example, the images captured at night can appear simultaneously on training and test set, making it harder to evaluate the algorithms' OOD generalization abilities on night images. Compared with [3], we consider the object detection task instead of the commonly used image classification task in OOD generalization algorithm evaluations.
>
> **1-2: Ctrlshift only has two-dimensional shift.**
>
> According to our theoretical analysis, the distribution shifts can be intrinsically categorized into correlation shift and diversity shift. One shift is complementary to the other shift in the mathematical sense, where attribute shift measures the spurious correlation in the same environment and diversity shift measures the environment changes. The main purpose of CtrlShift is to provide researchers with an easy-to-use dataset to evaluate algorithms on two-dimensional distribution shifts before conducting time-consuming experiments on large datasets. Other changes, such as Sim2Real, Weather, Scene and Time should belong to one kind of distribution shift. The algorithms' performance trend over the same kind of distribution shift should be similar.
>
> **2-1: More discussions on the correlation between ID and OOD performance, different generalization behaviors under different shifts, how the model architecture would affect ID and OOD performance.**
>
> Thanks for your constructive comments. We have added further discussion in Appendix A.5 due to the page limit. We will consider moving it to the main pages later.
>
> 1. High accuracy in IID may be the result of over-fitting since the spurious correlation exists in both the training and the testing distribution. Recently proposed methods, such as Deformable DETR, Sparse R-CNN and Swin Transformer, significantly outperform classic Faster R-CNN on IID datasets, however, have similar performance on OOD datasets. We can conclude that these methods make a prediction based on spurious correlated representations that existed in IID data and not existed in OOD data.
> 2. Defined by our theory in Appendix A.1, diversity shift is embodied by the novel features not shared by environments while correlation shift is caused by the shared non-causal features. Experiment results in Section 5.2 show that VREx and IRM obtain better OOD generalization ability against correlation shift than diversity shift, and it is a challenge to tackle diversity shift.
> 3. Current study starts to explore the impact of model architectures on OOD performance. NAS-OoD [6] applies a neural architecture search strategy to find the architecture with optimal OOD generalization ability and the results demonstrate that model architecture can significantly influence OOD accuracy. Some detectors have a relatively consistent performance on OOD, such as VarifocalNet, the proposed varifocal loss [7] can be considered to improve OOD generalization ability.
>
> **2-2: More importantly, there generally miss a good justification that DetectBench is a qualified benchmark that allows cross-compare different methods. Especially, it is already shown by many previous works that current object detectors fail at generalizing to OOD scenes [1] [2] [3].**
>
> We are the first to propose an OOD object detection benchmark for OOD algorithms (domain generalization algorithms), which are developed on classification scenarios. And our purpose is to provide a foothold and raise the attention of the OOD research community to turn to focus on more complex tasks.
>
> **3-1: The compared OOD generalization baselines are not adapted to the task of object detection.**
>
> This is one of the key pieces of information we want to deliver to the community of OOD generalization research in this paper. While most methods aim to be general OOD generalization methods, the evaluation is almost purely image classification tasks. This raises questions about recently proposed OOD methods' generalization abilities to other tasks beyond image classifications. Besides, as robust object detection is one of the key technologies for safer autonomous driving which may save people’s lives, we hope this work to be a foothold for future work on more general OOD generalization algorithms that are also capable of object detection and beyond, which is an important task in practical computer vision.

---

> > ### Comment · Reviewer_Sn2d · 2022-11-21
> > **Post rebuttal update**
> >
> > Thanks for the detailed feedback. I still have some follow up questions:
> >
> > I am still not quite convinced by the current justification on OOD-ODBENCH is a novel benchmark for domain generalization investigation on object detection.
> >
> > a) For instance, the authors argued in their 1-1 about how data splits would matter. Then, this should be very an important aspect of this paper to discuss how the proposed splits and why such splits make more sense, considering the authors mainly use existing datasets rather than propose new datasets. Proposing some splits and evaluating on a set of methods are not enough in my opinion.
> >
> > b) However, these syn2real, day2night, and weather scenarios are quite common cross-domain setups. The existing datasets provide the meta information to generate the splits. So, the add-on value of this paper is rather limited, as it does not really provide some novel splits beyond common practices and requires little extra effort to have them. For instance, in https://openaccess.thecvf.com/content/CVPR2022/papers/Wu_Single-Domain_Generalized_Object_Detection_in_Urban_Scene_via_Cyclic-Disentangled_Self-Distillation_CVPR_2022_paper.pdf, the authors already used the day2night of BDD100k for evaluation and their focus was on object detection with an improved domain generalization.
> >
> > 2. I still find quite hard to extract information from the paper. For instance, Table 3, now the authors added MS COCO as described in their answer 5. But I am not sure MS COCO is the ID for any of the other 4 columns, i.e., Sim2real, Weather, Scene, Time. I understand different scenarios have different source/target domains, then why not split into separate tables...If you dont show the ID performance, how can we interpret the OOD performance gap? For instance, a bad OOD performance could be: a) it is just an out-dated model which also performs bad in ID, or b) it performs very well on ID, but suffers a lot on OOD. Essentially, it is important to check the OOD and ID performance gap.

---

> > > ### Author Response · Authors · 2022-11-22
> > > **3.Response to Reviewer Sn2d**
> > >
> > > Thanks for your reply. We would like to re-emphasize that we are the first to evaluate OOD algorithms, which were mainly evaluated on classification tasks, on more practical task, such as object detection. Our findings suggest a performance gap between OOD classification and OOD object detection though a lot of OOD algorithms claim to be general. The main point of this paper is to raise concerns in OOD generalization research community to systematically investigate OOD generalization algorithms that are generally effective. The benchmarks provided in the paper can serve as a foothold for researchers to develop future OOD generalization algorithms.
> > >
> > > **C1: a) and b) Train-test split is not novel comparing to previous works**
> > >
> > > Previous train-test split can be summarized as "set2set" and "single domain". As we explain in 1-1 in 1.Response to Reviewer Sn2d, "set2set" may occur information overlap and this is a violation to OOD setting. As for "single domain"[1], previous domain generalization works [2-5] have already demonstrated that one of the keys to OOD generalization is the multi-domain training set, which is also the condition of most OOD algorithms [2-5], and we are the first to construct the multi-domain setting for evaluating OOD generalization algorithms on object detection.
> > >
> > > **C2: IID results**
> > >
> > > We provide the IID results of both MS COCO and BDD100K in the following table, where BDD100K(OOD) is the average accuracy on Weather, Scene and Time. BDD100K(IID) is the result on the original train-test split provided by the authors. As the results show, almost all the object detectors suffer from OOD conditions. Since BDD100K is a large-scale autonomous driving dataset and more complex than MS COCO, the degeneration from MS COCO to BDD100K(OOD) is much larger than from BDD100K(IID) to BDD100K(OOD). Due to the manuscript revision deadline, we will put these results in the later version.
> > >
> > > |Detector|MS COCO|BDD(IID)|BDD(OOD)|BDD(OOD)-MS COCO|BDD(OOD)-BDD(IID)|
> > > | :----- | :----:| :----:| :----: |:----:| :----: |
> > > |Faster R-CNN|42.1|34.0|32.6|-9.5|-1.4|
> > > |RetinaNet|41.0|34.3|31.7|-9.3|-2.6|
> > > |Mask R-CNN|42.8|34.8|32.5|-10.3|-2.3|
> > > |CornetNet|41.2|29.4|27.7|-13.5|-1.7|
> > > |YOLOv3|33.7|27.4|24.8|-8.9|-2.6|
> > > |FCOS|42.6|34.6|32.3|-10.3|-2.3|
> > > |Cascade R-CNN|44.7|35.4|32.4|-12.3|-3.0|
> > > |MS R-CNN|43.0|34.7|32.2|-10.8|-2.5|
> > > |Libra R-CNN|42.7|34.1|32.0|-10.7|-2.1|
> > > |Double-Head R-CNN|40.0|34.0|32.0|-8.0|-2.0|
> > > |VarifocalNet|50.4|37.0|35.0|-15.4|-2.0|
> > > |Sparse R-CNN|46.2|33.8|33.6|-12.6|-0.2|
> > > |DETR|42.0|25.7|21.2|-20.8|-4.5|
> > > |Deformable DETR|46.8|33.1|32.2|-14.6|-0.9|
> > > |Swin Transformer|51.9|35.8|32.5|-19.4|-3.3|
> > > |YOLOX|50.9|32.2|32.4|-18.5|+0.2|
> > >
> > >
> > > [1] Single-Domain Generalized Object Detection in Urban Scene
> > > via Cyclic-Disentangled Self-Distillation
> > > [2] Invariant Risk Minimization
> > > [3] Distributionally Robust Neural Networks for Group Shifts: On the Importance of Regularization for Worst-Case Generalization
> > > [4] Domain Generalization with Adversarial Feature Learning
> > > [5] Deep CORAL: Correlation Alignment for Deep Domain Adaptation

---

> > > > ### Comment · Reviewer_Sn2d · 2022-11-23
> > > > **OOD-CV benchmark and final comment**
> > > >
> > > > If we dont limit to driving scene datasets, I believe https://arxiv.org/pdf/2111.14341.pdf is a very relevant prior work and it also opened a OOD CV challenge on multiple tasks, including object detection, in ECCV2022.
> > > >
> > > > While OOD-CV has its own limitations, I think the discussion in the paper is richer, and also we can understand better what are harder shifts to address. For instance, as now shown in Table 3, YOLOX does not suffer from the gap. But if the discussion is simply limited to "almost all the object detectors suffer from OOD conditions", I am afraid the information is rather limited.
> > > >
> > > > OOD is a general problem for data-driven learning problems, and a major challenge. It is quite expected object detection will not be a special case. Putting OOD aside, it is quite clear that we cannot take it for granted that techniques would work for both image classification and object detection, as they are different tasks. Therefore, it is quite necessary to put more insightful discussion and data curations for such benchmark paper. As I also mentioned in my first comments, I think the paper targets important problem, nevertheless, the current content and presentation/writing are not so ready yet for acceptance in my opinion.

---

> > > > > ### Author Response · Authors · 2022-11-25
> > > > > **4.Response to Reviewer Sn2d**
> > > > >
> > > > > Thanks a lot for your valuable time. The main purpose of the paper is to raise concerns in the OOD generalization research community about the applicability of the algorithms beyond classification. To some extent, the gap between classification and object detection may result in the degeneration of algorithms' performance, however, what we expect is that the improvement compared to baseline (ERM) should remain between different tasks and the results show ERM performs the best among our benchmark datasets though most works claim they achieve SOTA OOD generalization ability. Domain generalization has been proposed for years yet few works focus beyond the Utopian condition (classification) and we hope this work can orient the current research to developing more general algorithms. Moreover, compared to existing benchmarks, we are the first to propose the multi-domain train-test split, which is the problem setting of domain generalization algorithms, and evaluate OOD algorithms on object detection. Thanks again for this constructive discussion.

---

> > > > > > ### Comment · Reviewer_Sn2d · 2022-11-25
> > > > > > **domain generalization on object detection is not new**
> > > > > >
> > > > > > Just to make it clear:
> > > > > > 1. people have been working on domain generalization on object detection, for instance, the CVPR paper I mentioned before.
> > > > > > 2. OOD benchmark in object detection also exists, e.g., OOD-CV (which also mentioned before)
> > > > > > 3. The train-test splits proposed in this work, some of them have already been considered in domain generalization paper, for instance, the CVPR paper used the day to night split of BDD.
> > > > > > 4. Multi-domain vs. Single-domain both are valid. For single-domain case, it is easier to see which scenario is more challenging. But, the authors mainly reported averaged performance across different train-test splits. This kind of information is also lost in the results.
> > > > > >
> > > > > > Thanks a lot for multiple iterations, and addressing my concerns. I do agree with the fact that in object detection domain generalization is less explored than in classification.
> > > > > >
> > > > > > I will keep my original rating.

---

> > > > > > > ### Author Response · Authors · 2022-11-26
> > > > > > > **5.Response to Reviewer Sn2d**
> > > > > > >
> > > > > > > Thanks a lot for making it clear.
> > > > > > >
> > > > > > > 1. Our effort is to benchmark this problem and serve as a foothold for future research. We do not consider the improvement of technology that can be quickly adapted to anything practical and related to the safety of human beings trivial, such as autonomous driving. If we only allow research based on some Utopian assumptions to be published and refused anything that might improve life-critical technology in the OOD research community. It will be hard for this research community to have impacts on industry.
> > > > > > >
> > > > > > > 2. OOD-CV does focus on the OOD problem, including object detection but, importantly, it has not considered the recently published OOD algorithms. We evaluate 12 OOD algorithms on 4 OOD benchmarks and we conclude that the current efforts on OOD are not efficient enough to outperform ERM on object detection, which we believe that it is due to the algorithms themselves rather than the difference between classification and object detection.
> > > > > > >
> > > > > > > 3. Day2night split is a single domain setting. Our benchmark uses a multi-domain split. We believe this attempt is critical for OOD generalization. Please refer to the following illustration.
> > > > > > >
> > > > > > > 4. Multi-domain is one of the most important keys to achieving OOD generalization ability. Consider the following example: If all the cars around the world are red, even for human beings, we will have the idea that cars are something red with four wheels. The reason why we do not consider color as an identification of cars is there are cars in different colors. And this is the essence of multi-domain. To achieve OOD generalization, models need to identify the invariant representations of objects, which are shared by different domains. If we do not have access to pre-trained knowledge, which may include the prior knowledge of objects, it is impossible to extract such invariant information using single domain since every common information within the single domain can be invariant. And as our previous response pointed out, the importance of multi-domain has been recognized by plenty of OOD research [1-4]. Therefore, we believe our attempt, which has not been cherished by previous works on object detection, should be novel even though it seems like a trivial change and it is sufficient enough to develop more practical and life-critical algorithms based on this foothold.
> > > > > > >
> > > > > > > [1] Invariant Risk Minimization
> > > > > > >
> > > > > > > [2] Distributionally Robust Neural Networks for Group Shifts: On the Importance of Regularization for Worst-Case Generalization
> > > > > > >
> > > > > > > [3] Domain Generalization with Adversarial Feature Learning
> > > > > > >
> > > > > > > [4] Deep CORAL: Correlation Alignment for Deep Domain Adaptation

---

### Official Review · Reviewer_m6Le · 2022-10-25

**Confidence:** 4
**Correctness:** 4
**Technical Novelty And Significance:** 4
**Empirical Novelty And Significance:** 4
**Recommendation:** 8

**Clarity, Quality, Novelty And Reproducibility:**

To me this paper is clear, the writing and presentation of results is good, results and conclusions are clear and supported by the results.

About quality, I think this is a high quality paper, it has clear contributions, a clear problem and solution, and major insights that can influence the state of the art in OOD-OD and particular for object detector design. A lot of the performance improvements in object detection are only in in-distribution datasets, and this benchmark would help researchers also evaluate improvements in OOD generalization performance.

I believe this paper to be novel, there is no benchmark for the OOD-OD generalization task, and as the authors mention, OOD generalization algorithms are not usually tested in object detectors and that makes them fail. This is very clear to me.


**Strength And Weaknesses:**


Strengths
- The paper is very well written and it is clear, ideas flow clearly, I have no further remarks about writing.
- An OOD generalization benchmark for object detection makes sense, as there is no dataset/benchmark specifically for this task, and the paper evaluation is of good quality, by combining standard object detectors with OOD generalization algorithms, to see if they work.
- It is a good idea to have test OOD generalization for object detection, as the authors argue, OOD generalization algorithms are not usually evaluated for more complex tasks like object detection, they usually are only evaluated for classification datasets.
- The proposed benchmark uses a selection of public datasets, covering multiple kinds of shifts, and additionally a synthetic dataset (CtrlShift) is proposed which has diversity and correlation shifts made in a controlled way, allowing the evaluation of the effect of both shifts combined.
- The design of the DetectBench benchmark seems sensible, it contains multiple sub-benchmarks where train and test sets contain non-intersecting sets of conditions, like weather, scene types, time, and simulation/reality, which make a simple test set that is out of distribution with respect of the training set.
- I believe that the evaluation and initial benchmark results presented in the paper are correct. There is a good selection of OOD generalization algorithms to test, and of object detectors (Faster R-CNN, DETR, RetinaNet) for the main results, which makes the overall conclusions robust.
- Two major conclusions of this benchmark are that improvements to OOD generalization in classification settings do not transfer to improvements to OOD generalization in object detection, as tested on Faster R-CNN like on Table 4, and on sim2real for Faster R-CNN, DETR, and RetinaNet, and measured by mAP in the OOD setting. In some cases the performance actually decreases instead of increase, and that in the OOD setting, diversity shift seems to affect OOD generalization algorithms much more than attribute shift. These two conclusions are good contributions to the state of the art.
- While the main results (Table 4) are presented for Faster R-CNN on the whole DetectBench benchmark, there are additional results on the sim2real sub-benchmark using Faster R-CNN, DETR, and RetinaNet about OOD generalization algorithms, producing a more robust conclusion that OOD generalization algorithms fail to improve performance in object detectors.
- The design of CtrlShift, while simple, seems to do the job, as Figure 3 shows, there is correlation shift produced by changing the color of a car model, and for diversity shift there is simulated snow added to the environment, and both shifts can be combined to study their joint effect.

Weaknesses
- I think the only weakness in this paper is the name of the Benchmark, DetectBench is very generic, sounds like any detection benchmark, it should have OOD and generalization (or similar abbreviations) in the name, to distinguish it from OOD detection benchmarks and standard OOD generalization benchmarks. Please find a good name that includes all of this.

Minor Issues
- Figure 5 and related heatmaps would be better presented with the axis labeled in the figure itself, instead of relying on the caption. Just as Figure 3, Figure 5 should have axis labels for diversity and correlation shift. Additionally to this, Figure 5 should have the origin at the bottom left instead the top left, which would make the figure easier to interpret.
- In Table 3, I think it would be good to also provide the original mAP of these detectors, in order to put the benchmarked numbers into context, so the reader can easily see if mAP is lower than expected or not.
- I believe that in Table 2, simulation and reality are flipped in the sim2real description (last row, third column).
- Could you provide a bit more information on CtrlShift? I see that there are 2000 images, but how are these distributed according to the two kinds of shifts?

**Summary Of The Paper:**

This paper is a benchmark for out of distribution generalization in object detection. The authors create a benchmark to evaluate out of distribution generalization algorithms specifically for the task of object detection (object classification and bounding box regression), using multiple well known datasets in specific settings, while also introducing a new synthetic dataset that can be used to study correlation and diversity shift.

The contributions are:
- A new benchmark DetectBench to evaluate OOD generalization in object detectors (authors call this OOD-OD).
- A new dataset CtrlShift to study diversity and attribute shift in OOD-OD.
- A good evaluation and important insights on the effect of diversity and attribute shift on OOD generalization algorithms in object detectors, signaling the object detectors are more affected by diversity shift than attribute shift. This is an important conclusion that can guide future object detector and OOD-OD method design to improve OOD generalization performance.
- The evaluation also indicates that standard OOD generalization algorithms (designed for classification), do not transfer successfully to object detection, indicating the need for more specialized OOD generalization algorithms for this task.

**Summary Of The Review:**

I believe that this paper should be accepted. There are only minor issues, there is a clear gap in the state of the art that this paper fills, a benchmark/dataset for out of distribution generalization for object detection tasks, and the paper does an excellent initial evaluation. This paper has good insights for future research, mainly that OOD generalization algorithms designed for classification fail to improve performance in object detection, and that object detectors are more sensitive to diversity than correlation shifts.

---

> ### Author Response · Authors · 2022-11-14
> **Response to Reviewer m6Le**
>
> Thank you very much for your positive comments. We have corrected the papers according to your advice. The corrected parts are shown in Red.
>
> **C1: Better name for the benchmark**
>
> Thank you very much for your constructive advice. We have changed the name to OOD-ODBench (Out-of-Distribution generalization Object Detection Benchmarks) to distinguish our work from general object detection benchmarks and OOD image classification benchmarks.
>
> **C2: Original mAP of detectors.**
>
> Thank you for your suggestions. We have added the experimental results of detectors on MS COCO as the original mAP of detectors in Table 3. Please refer to the revised version.
>
> **C3: Typo.**
>
> Thanks for pointing that out. We have revised the order in Table 2.
>
> **C4: More information about CtrlShift about the distributions according to the two kinds of shifts.**
>
> There are 109 white car images ($w$) , 52 red car images ($r$) and 40 blue car images ($b$) in CtrlShift. For correlation shift, given the quantity ratio $\rho_{cor}$, the training set is {$\rho_{cor} \cdot w$ , $(1-\rho_{cor}) \cdot r$ , $(1-\rho_{cor}) \cdot b$} and the testing set is  {0 , $\rho_{cor} \cdot r$ , $\rho_{cor} \cdot b$}. For diversity shift, we add snow weather effects on the testing set with the certain intensity $\rho_{div}$.

---

### Official Review · Reviewer_JACh · 2022-11-04

**Confidence:** 4
**Correctness:** 3
**Technical Novelty And Significance:** 3
**Empirical Novelty And Significance:** 2
**Recommendation:** 6

**Clarity, Quality, Novelty And Reproducibility:**

clarity/quality:
- The paper is well-written and easy to follow. The tables are clear to compare different set-ups and results

novelty:
- Good. Most of the OOD studies are on classification problems. And this work shows the OOD method is not always working on object detection tasks.

reproducibility:
- The author provides their code with a short readme file. Each experiment requires 8 V100 GPUs, which should be reproducible in the laboratory.

**Strength And Weaknesses:**

[Strength]
1. This paper works on an interesting and practical problem of OOD generalization of object detection models. The train-test discrepancy exists in many CV problems, and object detection has been well-studied and applied in research and industry. This DetectBench can work as a fundamental benchmark to study the OOD problem in object detection.

2. The benchmark covers multiple conditions in train-test discrepancy: weather, scene, time, and simulation. They can fairly work together to evaluate the OOD progress

3. Extensive experiments of 16 detectors and 12 OOD algorithms are measured on DetectBench. Those results are promising for the next generation of OOD-OD research.

4. The author has several new assumptions from the empirical studies. For example, they claim that improvement in object detection does not always transfer to OOD object detection, and the success in OOD is also not always consistent between classification and object detection,

[Weakness]
1. The technical contribution is relatively limited, for example, a better OOD-OD method can be proposed/suggested given the benchmark result.

2. Although it is sufficient to have more than ten OOD algorithms to draw the conclusion, but the re-implementation of a different problem may affect the observation. For example, using only 3 possible values for the hyper-parameter in all the methods may not be enough.

3. The synthetic CtrlShift dataset is very fine-grained on a single object class with a single instance per image, therefore, it is lack of generalization. A better motivation is also supposed to explain the importance of the 'car' class in object detection, and why car color and snow level are selected for the only domain shift options.




**Summary Of The Paper:**

This paper works as a benchmark of out-of-distribution object detection (OOD-OD), which evaluates the detector's generalization ability to a different data (open-world) domain. While many OOD methods have been proposed to amortize the domain gap, DetectBench, consisting of four datasets, shows the current OOD generalization algorithms no longer work on complex object detection tasks. This paper leads to an in-depth discussion on the progress of OOD work, and provides a foundation for OOD-OD research.

**Summary Of The Review:**

This work is a benchmark of out-of-distribution object detection. Although it has limited technique contribution, it is marginally above the threshold because the problem is novel and less studied.

---

> ### Author Response · Authors · 2022-11-14
> **Response to Reviewer JACh**
>
> **C1: The technical contribution is relatively limited**
>
> The main point of the paper is to provide a new perspective on evaluating OOD generalization algorithms. Instead of only focusing on image classification problems, object detection is a practical and largely untouched topic in OOD generalization research. Our works reveal that the high accuracy achieved by existing OOD generalization algorithms on image classification tasks is not easily transferred to object detection tasks, though many claimed to be general algorithms. Moreover, we discuss some critical issues for improving OOD generalization ability in Appendix A.5 in the revised version, including how the model architecture would affect IID and OOD performance.
>
> **C2: Only 3 possible values for the hyper-parameter may not be enough**
>
> Compared with (OOD) image classification dataset, object detection datasets contain a lot more image samples. It is strenuous and time-consuming to train object detection algorithms. Thus, object detection algorithms' performances cannot rely on the selection of hyper-parameters too much to be practical. In our experiments, we choose these three parameters to represent the slight, medium and heavy regularization, respectively, which is already able to show the trend of models' performance when we increase the penalty introduced by OOD generalization algorithms. Besides, due to the large evaluation data volume, the evaluation results should be relatively stable against small changes in hyper-parameters.  We can also report the results with more hyper-parameters upon request.
>
> **C3: CtrlShift is lack generalization.**
>
> According to our theoretical analysis, distribution shifts can be intrinsically categorized into correlation shift and diversity shift. The main purpose of CtrlShift is to provide researchers with an easy-to-use dataset to evaluate algorithms on two-dimensional distribution shifts before conducting time-consuming experiments on large datasets. Other changes, such as Sim2Real, Weather, Scene and Time should belong to one kind of distribution shift according to the definition of correlation shift and diversity shift. The algorithms' performance trend over the same kind of distribution shift should be similar.

---

### Author Response · Authors · 2022-11-14
**Summarization**

Thanks to all reviewers for their valuable time devoted to reviewing this paper.  In this work, we propose an OOD object detection benchmark for OOD algorithms (domain generalization algorithms that were mainly evaluated on (toy) image classification tasks. It is found that the huge improvements of recent OOD generalization algorithms reported on image classification tasks fail to transfer to OOD object detection tasks, though many claimed to be a general algorithm. Besides, recent progress on IID object detection tasks is not so prominent on OOD object detection tasks. In this paper, we hope to draw researchers’ attention to make a step forward in marrying important topics of OOD generalization and other practical tasks, such as object detection together, since this is one of the keys to safer autonomous driving which could be life-saving.

We are glad that Reviewer JACh finds this work interesting and practical with experiments conducted on 16 detectors, 12 OOD algorithms and 4 challenging OOD datasets. Some concerns were raised about the experimental setting and we have responded to the concerns one-to-one. We also appreciate the positive reviews given by Reviewer m6Le and revise the minor details according to his(or her) comments. Reviewer Sn2d considers this paper targets the important and unexplored OOD object detection problem and gives several constructive pieces of advice. According to the comments, we added the mAP results of detectors on MS COCO as the original performance of each detector in Table 3 to better support our main claims and added further discussion about several critical issues for improving OOD generalization ability in object detection. Reviewer aJ6J believes our work is well-motivated and thinks some experiments should be added to better support our main claims. Accordingly, we have provided the experimental results upon request.

We hope our responses to the reviewers can address their concerns well and hope that this benchmark can serve as a foothold for future OOD object detection research and raise the attention of the OOD research community on this challenging and practical problem.

---

### Decision · Program_Chairs · 2023-01-20

**Decision:**

Reject

**Justification For Why Not Higher Score:**

The issues with similarity of the proposed benchmark to others and lack of analysis of localization are significant and need to be throughly addressed to warrant publication.

**Justification For Why Not Lower Score:**

N/A

**Metareview: Summary, Strengths And Weaknesses:**

Paper proposes a new benchmark for out-of-distribution object detection (OOD-OD), which evaluates the detector's ability to generalize to new domains with changing data distribution. The paper was reviewed by four reviewers, ultimately receiving split scores:

- 1 x  marginally above the acceptance threshold
- 1 x  accept, good paper
- 2 x reject, not good enough

The main concerns with the paper came down to a couple of core issues:

1. Limited technical contributions and novelty (JACh and aJ6J)
2. Similarity and lack of discussion regarding differences with existing benchmarks in this space (Sn2d and aJ6J)
3. Limited findings and lack of more through analysis; paper only focuses on classification and not localization branch (Sn2d and aJ6J)

The rebuttal provided by the authors did not do a lot in alleviating the aforementioned concerns. While lack of technical contributions [1] is reasonable for the dataset and benchmarking paper, significant concerns with [2] and [3] remain. AC has carefully considered these issues. Indeed the paper is not the first to propose OOD detection benchmark, as such the onus is on the authors to appropriately compare and differentiate their benchmark with others. This, however, was not effectively done either in the original submission, nor in the submitted revision. The papers suggest by reviewers remain un-cited and there appears to be no discussion comparing settings, splits, and findings in this paper vs. predecessors; for example,

* Domain-Invariant Disentangled Network for Generalizable Object Detection
* Benchmarking Robustness in Object Detection: Autonomous Driving when Winter is Coming

While some of this discussion of this is given in rebuttal, this is insufficient and the paper itself needs to be adjusted and refocused to take these (and other provided) references into account. Further, AC agrees on the point made by Sn2d that analysis of object detection has to examine both classification and localization branch. For these reasons the decision is to Reject the paper at this time. Authors are encouraged to consider comments provided to them, revise the paper accordingly and, potentially, target another venue.